# Survey on Motion Planning for Multirotor Aerial Vehicles in Plan-Based Control Paradigm

**Geesara Kulathunga** [1,2] and **Alexandr Klimchik** [3,*]

1 Institute of Robotics and Computer Vision, Innopolis University, Innopolis 420500, Russia; g.mudiyanselage@innopolis.ru
2 Lincoln Institute for Agri-Food Tech, Lincoln Centre for Autonomous Systems, University of Lincoln, Riseholme Park, Lincoln LN2 2LG, UK
3 School of Computer Science, Lincoln Centre for Autonomous Systems, University of Lincoln, Lincoln LN6 7TS, UK
* Correspondence: aklimchik@lincoln.ac.uk

**Abstract:** In general, optimal motion planning can be performed both locally and globally. In such a planning, the choice in favor of either local or global planning technique mainly depends on whether the environmental conditions are dynamic or static. Hence, the most adequate choice is to use local planning or local planning alongside global planning. When designing optimal motion planning, both local and global, the key metrics to bear in mind are execution time, asymptotic optimality, and quick reaction to dynamic obstacles. Such planning approaches can address the aforementioned target metrics more efficiently compared to other approaches, such as path planning followed by smoothing. Thus, the foremost objective of this study is to analyze related literature in order to understand how the motion planning problem, especially the trajectory planning problem, is formulated when being applied for generating optimal trajectories in real-time for multirotor aerial vehicles, as well as how it impacts the listed metrics. As a result of this research, the trajectory planning problem was broken down into a set of subproblems, and the lists of methods for addressing each of the problems were identified and described in detail. Subsequently, the most prominent results from 2010 to 2022 were summarized and presented in the form of a timeline.

**Keywords:** multirotor aerial vehicles (MAVs); B-spline; minimum-snap; model predictive control (MPC); nonlinear model predictive control (NMPC); linear quadratic regulator (LQR); differential dynamic programming (DDP); optimal control problem (OCP); quadratic programming (QP); safe flight corridor (SFC); gradient-based trajectory optimization (GTO); truncated signed distance field (TSDF)

## 1. Introduction

Adroit motion planning of little flying creatures, such as birds and butterflies, is an extraordinarily demanding task for several reasons, including aggressive maneuvers. An example of such a high-speed maneuver need is one in particularly tight spots where the environment is obstacle-rich. Researchers have been trying to replicate similar maneuvers using two different types of aerial vehicles: conventional and unconventional. In this research, we deal with conventional aerial vehicles, for instance, unmanned aerial vehicles (UAVs), multirotor aerial vehicles (MAVs), etc. Recent progression in computation capabilities and embedded sensing has been boosting the procedure of mimicking natural flying animals; this advancement has enabled plenty of new opportunities in diverse fields: inspection, autonomous transportation, logistics, delivery, aerial photography, post-disaster, and medical services. However, optimal motion planning remains a crucial task in all the fields listed above. In optimal motion planning, the environmental reasoning cannot be predictable, since environmental conditions change rapidly. Hence, there are various challenges to be addressed to obtain highly efficient and optimal motion planning. In

this paper, we mainly focus on how researchers have been addressing these challenges in optimal motion planning to obtain robust navigation in various domains for multirotor aerial vehicles (MAVs).

In most industrial applications, the environment is either fully or partially unexplored, and unpredictable events can occur at any time due to a variety of factors. Therefore, a fast and accurate optimal motion planning technique is required to handle these unexpected problems in real time. The optimal motion planning problem is generally divided into three subcategories: path planning followed by smoothing, kinodynamic search-based trajectory generation, and motion-primitive-based approaches. Of these three, plan-based control approaches are the most widely used and efficient way to address the problem considered, compared to the other two approaches. Numerous plan-based control strategies have been proposed in the past decade, with promising results. This is one of the key motivating factors for reviewing plan-based control, especially for industrial multirotor aerial vehicles (MAVs). Most industrial multirotor aerial vehicles (MAVs), such as quadrotors, have low-level controllers, for example, PX4 [1], DJI [2], that can be operated independently irrespective of high-level execution commands. Moreover, such controllers reduce the overhead and complexity of developing high-level planning algorithms due to their independence. In other words, the same planner can be deployed on different firmware by implementing an interface between a high-level planner and a low-level controller. Thus, we narrowed down our study to considering only plan-based control approaches (Figure 1), particularly in application to industrial multirotor aerial vehicles (MAVs).

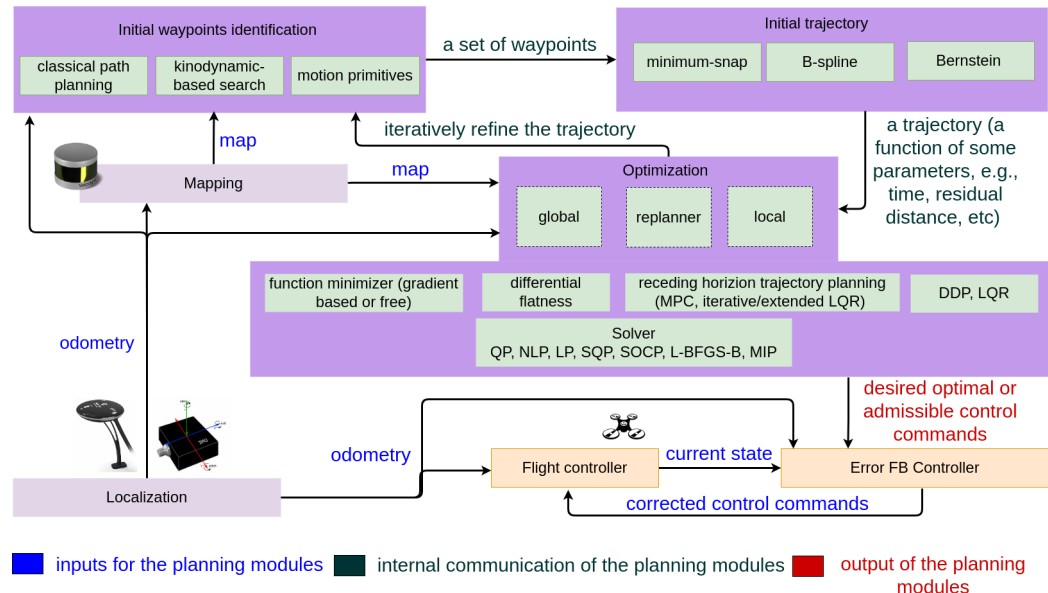

**Figure 1.** An overview of the plan-based control paradigm in the context of trajectory planning problem formulation. There are various ways to formulate the trajectory planning problem, each of which consists of a set of submodules (green color boxes) depending on the problem behavior.

The main limitation of multirotor aerial vehicles (MAVs) is low flight time. Hence, a multirotor aerial vehicle (MAV) should be capable of executing robust, agile, and aggressive maneuvers while ensuring dynamic feasibility and guaranteeing smoothness of the trajectory in low flight time. Furthermore, trajectory plotting should be performed within an obstacle-free zone at high speed to handle a given mission effectively. Such behavior is imposed by adhering to a set of constraints. If and only if the constraints are incorporated appropriately, desired needs can be fulfilled. Obtaining the right constraints at the right moment and applying appropriate control sequences to improve motion quality is the key objective of any plan-based control approach. However, the procedure of obtaining such right constraints is an open research problem due to its complexity and numerous other challenges that should be handled simultaneously. For example, the multirotor aerial

vehicle (MAV) has been widely employed in video-making-related fields in recent years, with cinematographic aerial shooting being one of the popular areas of interest during the last five years. In such shooting, generating smooth, obstacle-free trajectories is the main challenge. Various other challenges exist, and most of them are application-specific. In this work, we examine the most common problems related to trajectory planning applications in the paradigm of plan-based control, as well as how researchers have been alleviating those problems by proposing compelling solutions.

In optimal trajectory planning, trajectory generation and controlling the multirotor aerial vehicle are strongly interconnected. For multirotor aerial vehicles (MAVs), the trajectory generation process is relatively easy due to the dynamic properties of the multirotor aerial vehicles (MAVs). When dynamic obstacles are incorporated, the trajectory has to be refined at a high rate in order to keep a smooth maneuver despite increased computational demands. Moreover, understanding close-in obstacles' positions relative to the multirotor aerial vehicle (MAV) is crucial for making decisions in real time; this raises a new challenge: the one of the rapidity and accuracy of relative environment reconstruction, which essentially is how obstacle constraints are added to the problem formulation. Yet another challenge is of the impact of the obstacles and constraints on the smoothness and dynamic feasibility of the generated trajectory. After conducting an extensive literature review on the topic of trajectory planning for multirotor aerial vehicles (MAVs), we were able to isolate basic building blocks that are essential for optimal motion planning, as shown in Figure 2. Each of the primary components plays a key role in the process of trajectory generation.

- Convex segmentation: iterative regional inflation by semi-definite programming (IRIS), safe flight corridor (SFC), Stereographic Projection, Extracting convex polytopes
- Octomap and Euclidean signed distance field (ESDF) mapping, map building and construct KD-tree
- A set of geometrical shapes such as cubes, spheres and polyhedrons

- Differential Flatness and Partial Differential Flatness
- Empirical model
- Exact model

- iterative linear quadratic regulator (iLQR), Extended linear quadratic regulator (LQR), linear quadratic Gaussian (LQG), model predictive control (MPC), corridor-based model predictive contouring control (CMPCC)
- A set of control barrier functions (CBFs) for improving the robu

- Path planning, e.g., graph search techniques such as A-star and D-star, sampling-based techniques, i.e., RRT, RRT-star, rapidly exploring random graph (RRG)
- Kinodynamic and kinematic enable, A-star, RRT-star, FMT-star
- Incorporate motion primitive

Motion model selection

Free space segmentation

Receding horizon trajectory planning

Motion planning in plan-based control paradigm

Intermediate waypoints identification

- Refinement trajectory cost in most of the cases, defined by $J(\Gamma) = \xi_{smooth} J_{smooth}(\Gamma) + \xi_{obs} J_{obs}(\Gamma) + \xi_{soft} J_{soft}(\Gamma) + \xi_{end} J_{end}(\Gamma)$; different types of techniques are employed considering a few or all of the preceding individual costs: jerk or snap, end point, obstacle, elastic band for control points refinement

Continuous trajectory refinement

Initial trajectory generation

- Minimum-snap
- B-spline (uniform or nonuniform), Minimum-time B-spline
- Bernstein basis polynomial

**Figure 2.** The basic building blocks are encountered in trajectory planning problems. In general, a considered trajectory planning problem can be composed of one or more blocks sequentially or in parallel to fulfil the desired needs. Details explanation of each of the blocks are explained in the following sections: motion model selection (Section 2), intermediate waypoint identification (Section 3), initial trajectory generation (Section 4), free space segmentation (Section 5), continuous trajectory refinement (Section 6), and receding horizon trajectory planning (Section 7).

We survey the existing literature and identify the main building blocks of trajectory planning for MAVs: free-space segmentation, motion model selection, initial waypoint identification, initial trajectory generation, continuous trajectory refinement, and receding horizon trajectory planning. For each building block, we discuss and examine how previous research has addressed it. Furthermore, we used the same benchmark example to fairly compare different strategies. Also, we aim to provide a comprehensive overview of the recent advances and the most prominent results in trajectory planning for multirotor aerial vehicles (MAVs) from 2010 to 2022, with a focus on real-time generation of optimal trajectories as follows [3–57]:

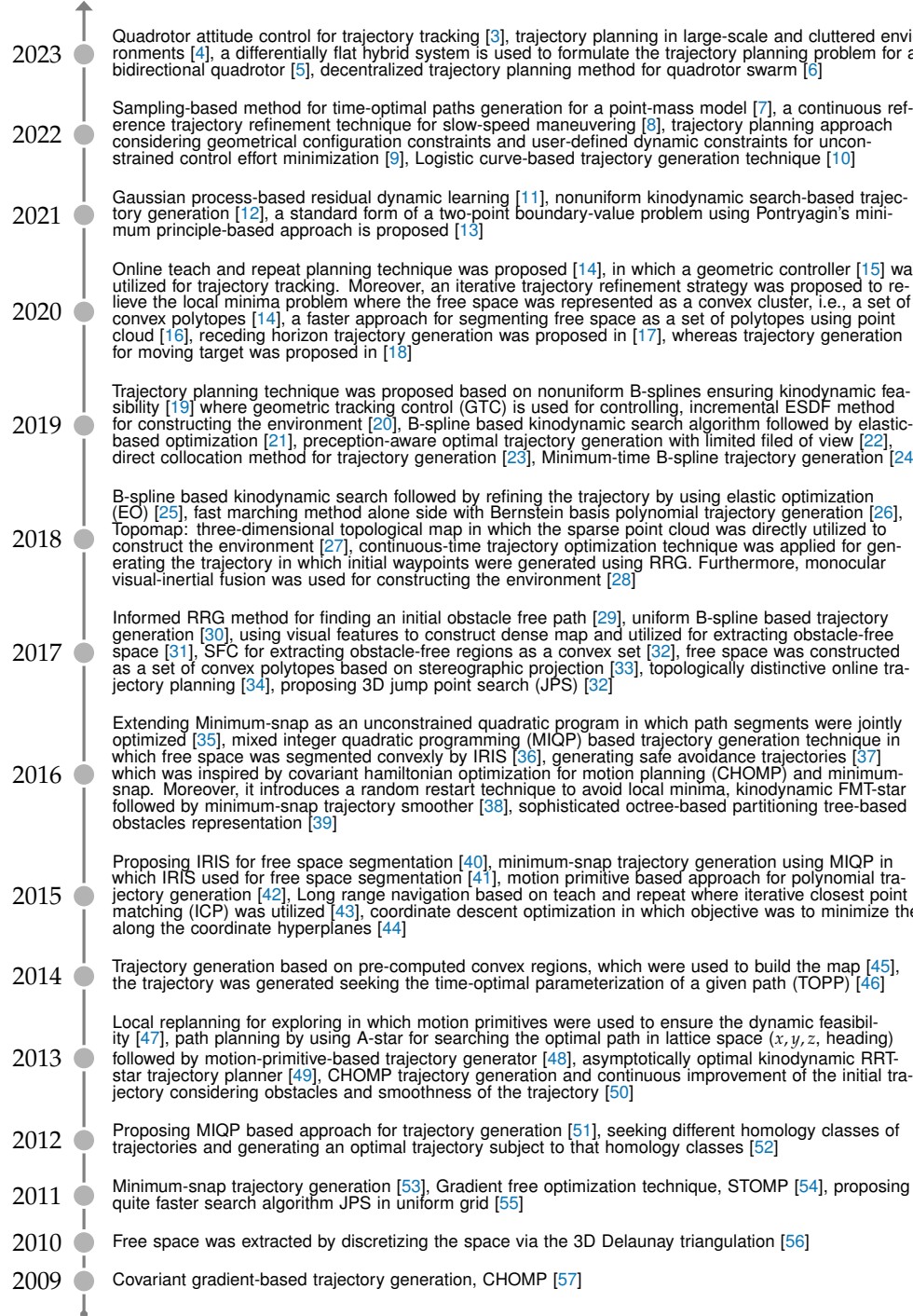

**2023** Quadrotor attitude control for trajectory tracking [3], trajectory planning in large-scale and cluttered environments [4], a differentially flat hybrid system is used to formulate the trajectory planning problem for a bidirectional quadrotor [5], decentralized trajectory planning method for quadrotor swarm [6]

**2022** Sampling-based method for time-optimal paths generation for a point-mass model [7], a continuous reference trajectory refinement technique for slow-speed maneuvering [8], trajectory planning approach considering geometrical configuration constraints and user-defined dynamic constraints for unconstrained control effort minimization [9], Logistic curve-based trajectory generation technique [10]

**2021** Gaussian process-based residual dynamic learning [11], nonuniform kinodynamic search-based trajectory generation [12], a standard form of a two-point boundary-value problem using Pontryagin's minimum principle-based approach is proposed [13]

**2020** Online teach and repeat planning technique was proposed [14], in which a geometric controller [15] was utilized for trajectory tracking. Moreover, an iterative trajectory refinement strategy was proposed to relieve the local minima problem where the free space was represented as a convex cluster, i.e., a set of convex polytopes [14], a faster approach for segmenting free space as a set of polytopes using point cloud [16], receding horizon trajectory generation was proposed in [17], whereas trajectory generation for moving target was proposed in [18]

**2019** Trajectory planning technique was proposed based on nonuniform B-splines ensuring kinodynamic feasibility [19] where geometric tracking control (GTC) is used for controlling, incremental ESDF method for constructing the environment [20], B-spline based kinodynamic search algorithm followed by elastic-based optimization [21], preception-aware optimal trajectory generation with limited filed of view [22], direct collocation method for trajectory generation [23], Minimum-time B-spline trajectory generation [24]

**2018** B-spline based kinodynamic search followed by refining the trajectory by using elastic optimization (EO) [25], fast marching method alone side with Bernstein basis polynomial trajectory generation [26], Topomap: three-dimensional topological map in which the sparse point cloud was directly utilized to construct the environment [27], continuous-time trajectory optimization technique was applied for generating the trajectory in which initial waypoints were generated using RRG. Furthermore, monocular visual-inertial fusion was used for constructing the environment [28]

**2017** Informed RRG method for finding an initial obstacle free path [29], uniform B-spline based trajectory generation [30], using visual features to construct dense map and utilized for extracting obstacle-free space [31], SFC for extracting obstacle-free regions as a convex set [32], free space was constructed as a set of convex polytopes based on stereographic projection [33], topologically distinctive online trajectory planning [34], proposing 3D jump point search (JPS) [32]

**2016** Extending Minimum-snap as an unconstrained quadratic program in which path segments were jointly optimized [35], mixed integer quadratic programming (MIQP) based trajectory generation technique in which free space was segmented convexly by IRIS [36], generating safe avoidance trajectories [37] which was inspired by covariant hamiltonian optimization for motion planning (CHOMP) and minimum-snap. Moreover, it introduces a random restart technique to avoid local minima, kinodynamic FMT-star followed by minimum-snap trajectory smoother [38], sophisticated octree-based partitioning tree-based obstacles representation [39]

**2015** Proposing IRIS for free space segmentation [40], minimum-snap trajectory generation using MIQP in which IRIS used for free space segmentation [41], motion primitive based approach for polynomial trajectory generation [42], Long range navigation based on teach and repeat where iterative closest point matching (ICP) was utilized [43], coordinate descent optimization in which objective was to minimize the along the coordinate hyperplanes [44]

**2014** Trajectory generation based on pre-computed convex regions, which were used to build the map [45], the trajectory was generated seeking the time-optimal parameterization of a given path (TOPP) [46]

**2013** Local replanning for exploring in which motion primitives were used to ensure the dynamic feasibility [47], path planning by using A-star for searching the optimal path in lattice space $(x, y, z, \text{heading})$ followed by motion-primitive-based trajectory generator [48], asymptotically optimal kinodynamic RRT-star trajectory planner [49], CHOMP trajectory generation and continuous improvement of the initial trajectory considering obstacles and smoothness of the trajectory [50]

**2012** Proposing MIQP based approach for trajectory generation [51], seeking different homology classes of trajectories and generating an optimal trajectory subject to that homology classes [52]

**2011** Minimum-snap trajectory generation [53], Gradient free optimization technique, STOMP [54], proposing quite faster search algorithm JPS in uniform grid [55]

**2010** Free space was extracted by discretizing the space via the 3D Delaunay triangulation [56]

**2009** Covariant gradient-based trajectory generation, CHOMP [57]

Our goal was to investigate trajectory planning for multirotor aerial vehicles (MAVs) in the plan-based control paradigm, focusing on analytical approaches rather than learning and evolutionary approaches. In future work, we plan to investigate trajectory planning

approaches based on imitation learning, inverse reinforcement learning, and evolutionary computing. Further investigations also include a comparison of different paradigms in a simulated environment and through real-world experiments for different operating scenarios: trajectory planning for high-dense and less-dense environments, static and dynamic obstacle avoidance, and high-speed and low-speed maneuvers. These considerations were outside of the scope of this work since we focused on the theoretical aspects of the plan-based trajectory planning paradigm. The rest of the paper is organized as follows:

- Section 2—Motion Modeling—discusses the type of motion model suitable for defining the dynamics of the robot based on the chosen trajectory generation technique. Exact, empirical, and differential flatness models are presented.
- Section 3—Initial Waypoint Generation—surveys state-of-the-art techniques for finding initial tentative waypoints for trajectory generation, focusing on graph search-based algorithms, motion-primitive-based approaches, and fast marching methods.
- Section 4—Initial Trajectory Generation—comprehensively reviews initial trajectory generation techniques. It begins by defining how to formulate a trajectory, followed by a detailed explanation of several interesting techniques, including minimum-snap, polynomial trajectory generation as quadratic programming (QP), unconstrained polynomial trajectory generation, covariant gradients, B-spline, and Bernstein. Finally, it compares several trajectory techniques to highlight their respective strengths and weaknesses.
- Section 5—Free Space Extraction—explains how to extract and incorporate free space into trajectory planning. OctoMap, IRIS, and SFC are the main methods discussed in this section.
- Section 6—Trajectory Refinement—describes the trajectory refinement process.
- Section 7—Horizon-Based Trajectory Planning—presents horizon-based trajectory planning techniques, starting with linear quadratic regulator (LQR) and its variants, such as iterative LQR (iLQR) and extended LQR (ELQR). It then covers advanced techniques, such as model predictive control (MPC) and its variants, including nonlinear MPC (NMPC).
- Section 8—Solvers for Optimization—details various solvers that can be used to solve the optimization problem, starting with quadratic programming formulation. It then lists and describes the usage of mixed-integer quadratic programming (MIQP), gradient-based trajectory optimization (GTO), BOBYQA, and many other solutions.

## 2. Motion Model Selection

Exact model, empirical model and differential flatness are the main techniques that can be employed for selecting the most appropriate motion model for a specified application. The appropriate motion model selection procedure varies depending on the problem formulation. For example, planning followed by controlling approaches does not necessarily have an exact motion model mainly due to high computational demands. In such scenarios, an empirical motion model is sufficient for planning, since a dedicated controller is utilized for controlling the quadrotor.

### 2.1. Exact Model

In general, multirotor aerial vehicle (MAV) dynamics is described by six degrees of freedom. However, in planning followed by high-level controlling approaches, it is not required to define an actual motion model for planning, since a high-level controller consists of a fully-fledged quadrotor motion model. In most circumstances, the planner is composed of approximated quadrotor dynamics; this is due to computational complexity, which is not adequate for real-time onboard processing. Hence, the motion model selection process depends on the approach that formulates needs. In [58], the researchers proposed a six-degrees-of-freedom motion model, whose state vector is defined by $\mathbf{x} = [\mathbf{p}^\top, \mathbf{v}^\top, \psi^\top, \omega^\top]$, where $\psi$, $\mathbf{p}$, $\mathbf{v}$ and $\omega$ stand for orientation (rad), position (m), velocity (m/s), and angular velocity (rad/s) in $\mathbb{R}^3$, respectively, with respect to a defined local coordinate

frame. The system input or total trust that is applied for each of the motors is given by $\mathbf{f} = [f_1, f_2, f_3, f_4]^T$ (N). System dynamics is determined as $\dot{\mathbf{x}} = [\dot{\mathbf{p}}^\top, \dot{\mathbf{v}}^\top, \dot{\psi}^\top, \dot{\omega}^\top]$, where $\dot{\mathbf{p}} = \mathbf{v}$, $\dot{\mathbf{v}} = -g \cdot \mathbf{e}_z + \frac{(\mathbf{f} \cdot \exp[\psi] \cdot \mathbf{e}_z - k_v \cdot \mathbf{v})}{m}$, $\dot{\psi} = \omega + \frac{1}{2}[\psi] \cdot \omega + (1 - \frac{1}{2}\frac{\|\psi\|}{tan(\frac{1}{2}\|\psi\|)})[\psi]^2 \cdot \omega / \|\psi\|^2$, $\dot{\omega} = J^{-1}(\rho(f_2 - f_4)\mathbf{e}_x) + \rho(f_3 - f_1)\mathbf{e}_y + k_m(f_1 - f_2 + f_3 - f_4)\mathbf{e}_z - [\omega] \cdot J \cdot w)$, $g = 9.8\,\mathrm{ms}^{-2}$ and $\mathbf{e}_i$, $i = x, y, z$ stand for standard basis vectors in $\mathbb{R}^3$, and $k_v, m, J, \rho$, and $k_m$ are robot-specific constants.

## 2.2. Empirical Model

Other than the exact model, a six-degree-of-freedom motion model was proposed for governing quadrotor in a distributed setup [59]. Later, it was reduced to four-degree-of-freedom motion model [60]. Furthermore, in [61], a 4-degree of freedom (DoF) motion was used for controlling several quadrotors in a distributed setup in which nonlinear model predictive control (NMPC) and model horizon estimation (MHE) are incorporated for relative tracking, where the relative motion model was defined as:

$$\dot{\mathbf{x}} = \mathbf{f}_c(\mathbf{x}, \mathbf{u}, \psi_z) = \begin{bmatrix} \dot{\mathbf{p}}_x \\ \dot{\mathbf{p}}_y \\ \dot{\mathbf{p}}_z \\ \dot{\psi}_z \end{bmatrix} = \begin{bmatrix} \mathbf{v}_x cos(\psi_z) - \mathbf{v}_y sin(\psi_z) - \bar{\mathbf{v}}_x + \mathbf{p}_y \bar{\dot{\psi}}_z \\ \mathbf{v}_x sin(\psi_z) + \mathbf{v}_y cos(\psi_z) - \bar{\mathbf{v}}_y - \mathbf{p}_x \bar{\dot{\psi}}_z \\ \mathbf{v}_z - \bar{\mathbf{v}}_z \\ \dot{\psi}_z - \bar{\dot{\psi}}_z \end{bmatrix}, \tag{1}$$

where the function $\mathbf{f}_c(\cdot) : \mathbb{R}^{n_u} \times \mathbb{R}^{n_x} \times \mathbb{R}^{n_{ru}} \to \mathbb{R}^{n_x}$ and $n_x = n_u = n_{ru} = 4$. The current control input is given by $\mathbf{u} = [\mathbf{v}_x, \mathbf{v}_y, \mathbf{v}_z, \dot{\psi}_z]$, whereas relative control input $\mathbf{u}_{ru}$ is denoted by $[\bar{\mathbf{v}}_x, \bar{\mathbf{v}}_y, \bar{\mathbf{v}}_z, \bar{\dot{\psi}}_z]$. $\mathbf{x} = [\mathbf{p}_x, \mathbf{p}_y, \mathbf{p}_z, \psi_z]$ is the state of the motion model, where $\mathbf{p}_i$, $i \in \{x, y, z\}$ is the position of the MAV in the world frame. $\psi_z$ and $\bar{\psi}_z$ denote the yaw angle or heading angle around the z axis and relative yaw angle, respectively. Derivative of $\psi_z$ and $\bar{\psi}_z$ are denoted by $\dot{\psi}_z$ and $\bar{\dot{\psi}}_z$, respectively. $\mathbf{v}_i$, $i \in \{x, y, z\}$ denote the velocities on each direction, whereas $\dot{\mathbf{p}}_i$, $i \in \{x, y, z\}$ gives the derivatives of $\mathbf{p}_i$. Since discrete space was chosen for controlling the system, Euler discrete model (1) was formulated as follows:

$$\mathbf{x}^+ = \mathbf{f}_d(\mathbf{x}, \mathbf{u}, \psi_z) = \begin{bmatrix} p_x \\ p_y \\ p_z \\ \psi_z \end{bmatrix} + \delta \begin{bmatrix} v_x cos(\psi_z) - v_y sin(\psi_z) - \bar{v}_x + y\bar{\dot{\psi}}_z \\ v_x sin(\psi_z) + v_y cos(\psi_z) - \bar{v}_y - x\bar{\dot{\psi}}_z \\ v_z - \bar{v}_z \\ \dot{\psi}_z - \bar{\dot{\psi}}_z \end{bmatrix}, \tag{2}$$

where $\delta$ is the sampling period and $\mathbf{f}_d(\cdot) : \mathbb{R}^{n_x} \times \mathbb{R}^{n_u} \times \mathbb{R}^{n_{ru}} \to \mathbb{R}^{n_x}$. $\mathbf{f}_c$ and $\mathbf{f}_d$ denote continuous and discrete dynamics, respectively. $\mathbf{x}^+$ depicts the next state given the current state $\mathbf{x}$. Subsequently, the motion model was simplified to 4-DoF for trajectory tracking for a quadrotor [62] (Equation (1)). In this trajectory-tracking approach, planning followed by the high-level controlling paradigm was applied. Such an approach was introduced because a simplified motion model is a reasonable choice for achieving real-time performance. Quadrotor state was defined as $\mathbf{x} = [\mathbf{p}_x, \mathbf{p}_y, \mathbf{p}_z, \psi_z]^T \in \mathbb{R}^{n_x}$, whereas input to the system was given by $\mathbf{u} = [\mathbf{v}_x, \mathbf{v}_y, \mathbf{v}_z, \dot{\psi}_z]^T \in \mathbb{R}^{n_u}$. The simplified motion model was given by:

$$\dot{\mathbf{x}} = \mathbf{f}_c(\mathbf{x}, \mathbf{u}) = \begin{bmatrix} \dot{\mathbf{p}}_x \\ \dot{\mathbf{p}}_y \\ \dot{\mathbf{p}}_z \\ \dot{\psi}_z \end{bmatrix} = \begin{bmatrix} \mathbf{v}_x cos(\psi_z) - \mathbf{v}_y sin(\psi_z) \\ \mathbf{v}_x sin(\psi_z) + \mathbf{v}_y cos(\psi_z) \\ \mathbf{v}_z \\ \dot{\psi}_z \end{bmatrix}, \tag{3}$$

where $\mathbf{f}_c(\cdot) : \mathbb{R}^{n_x} \times \mathbb{R}^{n_u} \to \mathbb{R}^{n_x}$ and $n_x = n_u = 4$. The discretization of (3) was given by:

$$\mathbf{x}^+ = \mathbf{f}_d(\mathbf{x}, \mathbf{u}) = \begin{bmatrix} \mathbf{p}_x \\ \mathbf{p}_y \\ \mathbf{p}_z \\ \psi_z \end{bmatrix} + \delta \begin{bmatrix} \mathbf{v}_x cos(\psi_z) - \mathbf{v}_y sin(\psi_z) \\ \mathbf{v}_x sin(\psi_z) + \mathbf{v}_y cos(\psi_z) \\ \mathbf{v}_z \\ \dot{\psi}_z \end{bmatrix}, \tag{4}$$

where $\mathbf{f_d}(\cdot) : \mathbb{R}^{n_x} \times \mathbb{R}^{n_u} \to \mathbb{R}^{n_x}$.

*2.3. Differential Flatness*

Here, differential flatness [63] provides algebraic functions (e.g., polynomials) which analytically map the trajectory and whose higher-order derivatives map to system states and inputs. Since the $N$th-order polynomial can be differentiated up to $N - 1$ times, the differential flatness property ensures the feasibility of the trajectory and generates appropriate control commands. More precisely, let the following:

$$\dot{\mathbf{x}} = \mathbf{f}_c(\mathbf{x}, \mathbf{u}) \quad \mathbf{x} \in \mathbb{R}^{n_x}, \mathbf{u} \in \mathbb{R}^{n_u}. \tag{5}$$

be a nonlinear system. According to to [64], if the system is differentially flat, there always exists a flat output, namely $\mathbf{z} \in \mathbb{R}^{n_z}$, where the dimension of the output is given by $n_z$. In such a system, states and control inputs can also be formulated from the system flat outputs whose derivatives are mapped through functions, namely $\varrho$ and $\tau$. Let $\mathbf{z} = \Im(\mathbf{x}, \mathbf{u}, \dot{\mathbf{u}}, \dots, \mathbf{u}^{(q)})$ be the flat output, holding $\mathbf{x} = \varrho(\mathbf{z}, \dot{\mathbf{z}}, \dots, \mathbf{z}^{(r)})$ and $\mathbf{u} = \tau(\mathbf{z}, \dot{\mathbf{z}}, \dots, \mathbf{z}^{(r)})$, where apices $^{(i)}$ stipulates the $i$th derivative. Along with that, the explicit trajectory generation process can benefit when it uses differentially flat systems, for example, $\varrho$ and $\tau$ can be a $d$th-order polynomial $p(t)$. Then, $\mathbf{x}^\top(t) = [p^\top(t) \ \dot{p}^\top(t) \ \ddot{p}^\top(t)]$ are the state of the system at time $t$ in which $\dot{p}^T$ and $\ddot{p}^T$ indicate the velocity and acceleration of the system, respectively. Control inputs can be determined by jerk [65], namely $\dddot{p}^T(t)$, where $p(t) = \lambda_d t^d + \dots + \lambda_1 t + \lambda_0, \ t \in [0, dt]$, where $\lambda_i, i = 0, \dots, d$ are the polynomial coefficients. There are various ways to construct these kinds of polynomials, including Minimum-snap, B-spline, etc.

## 3. Initial Waypoint Identification

Generally speaking, robots have a limited sensing range. Thus, planning a trajectory out of such a sensing range would be counterproductive. Hence, local trajectory planning and refinement when a robot moves is the optimal choice. With the help of sensing capabilities within the robots' sensing range, the robot's surrounded environment can be constructed as the intersection of three separate disjoint sets: free-known ($C_{free}$), occupied ($C_{obs}$), and unknown ($C_{unknown}$). Once $C_{free} \cup C_{unknown}$ is identified, a set of intermediate waypoints is needed to navigate the robot along the trajectory from the start position to the desired position. There are various techniques for finding a set of intermediate waypoints: sampling-based techniques (e.g., RRT*, probabilistic road map (PRM)) and path-searching techniques (e.g., A*, D*, JPS), where waypoint poses are given in the UTM (universal transverse mercator) coordinate system and then converted into the local coordinate system. Moreover, kinodynamic properties are incorporated into preceding intermediate waypoints finding techniques to ensure the dynamic feasibility of the robot. One of the first kinodynamic-based path planning approaches was proposed in [66], in which a variant of the A* method alongside with kinodynamic properties was applied to ensure the dynamic feasibility. Subsequently, several different methods were proposed for enhancing path planning, ensuring the dynamic feasibility by kinodynamic properties, including motion-primitive-based approaches.

Motion-primitive-based approaches [42,67,68] can be utilized for finding intermediate waypoints and for trxajectory generation. Gordon et al. [69] proposed a set of motion primitives for connecting edges of the graph that was constructed from A*. In this method, motion primitives were used to defining state vector $\mathbf{x}(t)$ and control input $\mathbf{u}(t)$ as a linear time invariant (LTI) system as follows:

$$\mathbf{x}_i(t) = [p_i(t)^\top, \dot{p}_i(t)^\top, \dots, p_i^{(k_r-1)}(t)^\top]^\top \in \mathbf{x}_i(t) \subset \mathbb{R}^{3 \times k_r},$$
$$p_i(t) = [p_x(t), p_y(t), p_z(t)]^T, \mathbf{u}_i(t) = p^{(k_r)}(t), \tag{6}$$

where $\mathbf{p}_\mu(t) = \Sigma_{j=0}^{d}\lambda_j t^j$, $\mu \in \{x, y, z\}$, which is formulated similar to (16), while $k_r$ and $d$ are the order of the derivative and the order of the polynomial, respectively.

$$\dot{\mathbf{x}}_i(t) = A\mathbf{x}_i(t) + B\mathbf{u}_i(t),$$

$$A = \begin{bmatrix} 0 & I_3 & 0 & \cdots & 0 \\ 0 & 0 & I_3 & \cdots & 0 \\ \vdots & \vdots & \vdots & \ddots & \vdots \\ 0 & \cdots & \cdots & 0 & I_3 \\ 0 & \cdots & \cdots & 0 & 0 \end{bmatrix}, \quad B = \begin{bmatrix} 0 \\ 0 \\ \vdots \\ 0 \\ I_3 \end{bmatrix}. \tag{7}$$

Hence, given control policy $\mathbf{u}_i(t)$ and initial state $\mathbf{x}(0)$, a sequence of succeeding states for a given time duration is determined by:

$$\mathbf{x}_i(t) = e^{At}\mathbf{x}(0) + \int_0^t e^{A(t-\gamma)}B\mathbf{u}(\gamma)d\gamma, \tag{8}$$

where $\gamma$ is the time duration that control policy is applied. In [69], to define the actual and heuristic cost of A*, the researchers used motion primitives, which are defined (as shown) in (8), and calculated initial waypoints set.

Another interesting approach to finding a set of initial intermediate waypoints is by using fast marching methods. In general, fast marching methods [70] are applied to track the propagation of a convoluted interface such as wavefront, especially in image processing. Let $\varphi$ be a close curve in a plane $\in \mathbb{R}^3$ that propagates orthogonally to the plane with a speed $v(\mathbf{p})$, assume $v > 0$. Given $\bigtriangledown T$ time period, propagation of the plane can be described by $|\bigtriangledown T(x)| = \frac{1}{v(\mathbf{p})}$ based on Eikonal partial differential Equation [71], where $\mathbf{p}$ is the position in $\mathbb{R}^3$ and the arrival time is formulated by $T(x)$. The fast marching concept was applied for path searching in [26] by proposing a method for calculating a velocity map. In this method, the arrival time was determined by assessing the desired velocity at the considered position in the local coordinate system. Hence, arrival time was calculated by backtracking from the goal pose to the start pose along the minimum cost path, which can be estimated from the gradient descendant. Though gradient descendant may trap in a local minimum, when smart marching is applied, gradient descendant does not trap in local minimum due to fast marching nature; this property was proven in [72]. To define the velocity map, ESDF was utilized to get the closest obstacle poses from the given pose. A quadrotor should move faster when there are no close-in obstacles and should be slower when it is moving through a cluttered environment. Such a behavior was mimicked by incorporating a hyperbolic tangential function, i.e., *tanh*. With such an assumption, the corresponding velocity was calculated based on (9):

$$v(l) = \begin{cases} v_{max}(tanh(l-e)+1)/2, & 0 \le l \\ 0, & l < 0' \end{cases} \tag{9}$$

where $v_{max}$ is the maximum velocity a quadrotor can fly, $l$ is the distance to the closest obstacle from the considered pose $\mathbf{p}$ and, $e$ is Euler's constant.

## 4. Initial Trajectory Generation

Let us consider a nonlinear system in the form of $\dot{\mathbf{x}}(t) = \mathbf{f}_c(\mathbf{x}(t), \mathbf{u}(t))$ with initial state $\mathbf{x}(t_0) = \mathbf{x}_0$, where state vector and control inputs are denoted by $\mathbf{x} \in R^{n_x}$ and $\mathbf{u} \in R^{n_u}$, respectively. When generating an initial trajectory ($\Gamma$), ensuring dynamic feasibility is a must. In other words, $\mathbf{x}$ and $\mathbf{u}$ satisfy the following constraints:

$$\mathbf{x} \in X \subseteq \mathbf{R}^{n_x}, \quad \mathbf{u} \in U \subseteq \mathbf{R}^{n_u}. \tag{10}$$

In addition to these constraints, safety constraints should also be imposed after reasoning the environment, to guarantee safety. The environment or configuration space $C$ can be

decomposed into $C_{obs}$ and $C_{free}$. Hence, a set of constraints should be introduced for the quadrotor to always be within free space $\mathbf{x} \in C_{free} = C / C_{obs}$. Hence, the initial trajectory generation process has to consider both said types of constraints simultaneously so that the quadrotor would have a smooth flying experience.

*4.1. Define Trajectory*

Let $\Gamma \leftarrow C \subset \mathbb{R}^d$ be an initial trajectory, which is parameterized as a function of time where $d$ denotes the $C$'s dimension. Since $\Gamma$ is a function, the objective of the trajectory generator is to determine the precise objective, which will eventually provide the optimal trajectory in a timely manner satisfying constraints and hypotheses that are imposed. Hence, optimal trajectory, namely $\Gamma^*$, can be posed as a discrete or continuous optimal control problem (OCP) [73]:

$$
\begin{aligned}
\Gamma^* = \min_{\mathbf{u}(\cdot)} \quad & J(\mathbf{x}(0), \mathbf{u}(\cdot)) \\
\text{s.t.} \quad & \mathbf{x}(0) = \mathbf{x}_0, \ \mathbf{x}(t_n) = \mathbf{x}_n \\
& \dot{\mathbf{x}}(t) = \mathbf{f}_c(\mathbf{x}(t), \mathbf{u}(t)) \\
& \mathbf{x}(t) \in C_{free}, \ \mathbf{u}(t) \in U, \ t \in [t_0, t_n],
\end{aligned}
\tag{11}
$$

where $t_0$ and $t_n$ denote the start and terminal time, respectively. Yet another challenging problem is to formulate the objective function, namely $J$. In the following subsections, we discuss several approaches to address this problem.

*4.2. Minimum-Snap-Based Trajectory Generation*

minimum-snap trajectory generation [53] uses the differential flatness property (Section 2.3) to automate the trajectory generation process. Let quadrotor trajectory be $\Gamma_T(t) = [r_T(t), \psi_T(t)]^T$ for flat output $[x, y, z, \psi_z]^T$, where $r = [x, y, z]$ is the center position of the MAV with respect to world coordinate system and $\psi_z$ is the yaw angle of the MAV. The continuous trajectory can be expressed as follows:

$$
\Gamma(t) : [t_0, t_n] \leftarrow \mathbb{R}^d,
\tag{12}
$$

where $d$ is the dimension of the space, e.g., 3. As we defined in Section 2.3, system states and inputs can be determined in terms of $\Gamma$ and its derivatives. $\Gamma, \dot{\Gamma}$, and $\ddot{\Gamma}$ correspond to position, velocity, and acceleration, respectively. Flat output and its derivatives estimation in minimum-snap refer to the original work [53] (Equations (1)–(35)).

In minimum-snap trajectory parameterization, the total time duration of the trajectory is divided into a set of subintervals, i.e., keyframes. Each keyframe consists of a desired position and a yaw angle. A safe corridor is constructed between consecutive keyframes as a set of piece-wise-polynomial functions to estimate smooth transitions through the keyframes. Let $m_d$ and $d$ be the number of keyframes and the order of the piece-wise-polynomial functions, respectively. Hence, $\Gamma_T(t)$ can be formulated as:

$$
\Gamma_T(t) = 
\begin{cases}
\Sigma_{i=0}^d \Gamma_{i,1}(t - t_0)^i & t_0 \le t < t_1 \\
\Sigma_{i=0}^d \Gamma_{i,2}(t - t_1)^i & t_1 \le t < t_2 \\
\quad \vdots & \\
\Sigma_{i=0}^d \Gamma_{i,m_d}(t - t_{m_d-1})^i & t_{m_d-1} \le t < t_{m_d}
\end{cases}.
\tag{13}
$$

To generate an optimal trajectory, the following objective is utilized:

$$J(r_T, \psi_T) = \int_{t_0}^{t_{m_d}} \xi_r \left\| \frac{d^{k_r} r_T}{dt^{k_r}} \right\|^2 dt + \xi_\psi \frac{d^{k_\psi} \psi_T}{dt^{k_\psi}}^2 dt$$

$$\min_w \quad J(r_T, \psi_T)$$

$$\text{s.t.} \quad \Gamma_T(t_i) = \Gamma_i \quad i = 1, \dots, m_d$$

$$\frac{d^p x_T}{dt^p}\Big|_{t=t_j} \leq 0 \quad j = 0, m_d; \ p = 1, \dots, k_r \tag{14}$$

$$\frac{d^p y_T}{dt^p}\Big|_{t=t_j} \leq 0 \quad j = 0, m_d; \ p = 1, \dots, k_r$$

$$\frac{d^p z_T}{dt^p}\Big|_{t=t_j} \leq 0 \quad j = 0, m_d; \ p = 1, \dots, k_r$$

$$\frac{d^p \psi_T}{dt^p}\Big|_{t=t_j} \leq 0 \quad j = 0, m_d; \ p = 1, \dots, k_\psi,$$

where $\xi_r$ and $\xi_\psi$ are regulation parameters, $k_r$ and $k_\psi$ are the order of derivation at each keyframe, and $\Gamma_T(t_i) = [x_i, y_i, z_i, \psi_{z_i}]^T, i = 0, \dots, T$. Time intervals, $t_1, t_2, \dots, t_{m_d}$ can be kept constant or varying when deriving the minimum-snap trajectory generation. In most cases, having varying time intervals between keyframes is necessary. Mellinger et al. [53] proposed a gradient descent-based approach for finding optimal time intervals between keyframes. Furthermore, Chen et al. [74] utilized A* to find the intermediate waypoints. Based on these estimations, time segments or keyframes are calculated incorporating both velocity and acceleration limits. In the latter approach, the steps listed below were used to obtain intermediate waypoints. Initially, the environment was constructed as a map using OctoMap. Afterwards, the formed map was split into two subsets: allocated and nonallocated (a set of free spaces). Then, the discrete graph was constructed connecting consecutive free spaces, which were represented as cubes. Afterwards, A* was applied for finding the optimal path segment within each cube. Similar to (14), the researchers set $k_r = 3$ and minimized only total jerk (15) to minimize the angular velocity. As an aside, minimizing the angular velocity helps to avoid fast rotation.

$$J = \int_{t_0}^{t_{m_d}} \xi_r \left\| \frac{d^{k_r} \Gamma_T(t)}{dt^{k_r}} \right\|^2 dt. \tag{15}$$

*4.3. Polynomial Trajectory Generation as QP*

In minimum-snap trajectory generation, total trust force, i.e., attitude acceleration, is proportional to the fourth derivative (snap) of the trajectory [53]. The gracefulness of such behavior helps to avoid generating excessive control commands. Subsequently, a slight variation of minimum-snap trajectory generation was proposed in [35], where segment times or keyframes were fixed initially. Once start and goal positions were provided, RRT* [49] was utilized for finding an obstacle-free path between the start and the goal poses as a sequence of optimal waypoints. Initial segment times ($m_d$), which were estimated using optimal waypoints, were calculated according to the maximum velocities that the quadrotor is allowed to fly due to set technical limits. Let $p_i(t)$ be the $d$th-order polynomial in the $i$th segment that describes as follows:

$$p_i(t) = \lambda_0 t^0 + \lambda_1 t^1 + \lambda_2 t^2 + \lambda_3 t^3 + \cdots + \lambda_d t^d. \tag{16}$$

Each $p_i(t)$ provides a flat output for a given time index $t$. $\lambda_j, j = 0, \ldots, d$ denotes the polynomial coefficients. The objective or cost function $J(\Gamma_i)$ can be fully determined by penalizing the derivatives of squares [35]:

$$J(\Gamma_i) = \int_{t_i}^{t_{i+1}} \xi_0 p_i(t)^2 + \xi_1 \dot{p}_i(t)^2 + \xi_2 \ddot{p}_i(t)^2 + \cdots + \xi_{k_r} p^{(k_r^i)}(t)^2 = P_i^T Q(T_i) P_i, \quad (17)$$

where $P_i$ is a vector whose elements contain polynomial coefficients: $\xi_0, \xi_1, \ldots, \xi_{k_r^i}$, where $k_r^i$ is the highest order of derivative and $Q(T_i)$ is Hassin matrix, which contains the $i$th segment squares of derivatives. Since there are $m_d$ number of segments, total cost $J(\Gamma)$ can be expressed by:

$$J(\Gamma) = \begin{bmatrix} P_1 \\ \vdots \\ P_{m_d} \end{bmatrix}^T \begin{bmatrix} Q(T_1) & & \\ & \ddots & \\ & & Q(T_{m_d}) \end{bmatrix} \begin{bmatrix} P_1 \\ \vdots \\ P_{m_d} \end{bmatrix}. \quad (18)$$

For a smooth flight experience, ensuring the continuity of derivatives between segments is necessary. Hence, imposing constraints between segments, e.g., velocity, acceleration, jerk, and snap is needed, which can be formulated as follows:

$$C_i p_i = \mathbf{d}_i, \quad C_i = \begin{bmatrix} \xi_0 \\ \xi_{k_r} \end{bmatrix}_i, \quad \mathbf{d}_i = \begin{bmatrix} d_0 \\ d_{k_r} \end{bmatrix}_i, \quad (19)$$

where $C_i$ contains a mapping matrix whose entries contain the start and end coefficients of $i$th segment, whereas $d_i$ contains derivative values, i.e., start and end of $i$th segment. Taking all constraints of $m_n$ segments:

$$C \begin{bmatrix} p_1 \\ \vdots \\ p_{m_d} \end{bmatrix} = \begin{bmatrix} \mathbf{d}_1 \\ \vdots \\ \mathbf{d}_{m_d} \end{bmatrix}. \quad (20)$$

Now this can be solved as a constrained quadratic programming (QP) problem.

### 4.4. Unconstrained Polynomial Trajectory Generation

The techniques that are used for uconstrained trajectory optimization are faster than constraints optimization. In [35], the researchers extended minimum-snap trajectory generation as an unconstrained QP. According to their findings, minimum-snap works well for small segments size. For higher-order polynomials with varying segment sizes, minimum-snap becomes ill-conditioned. Thus, an unconstrained QP was proposed. After substituting (19) and (20) into (18), $J(\Gamma)$ can be reformulated as:

$$J(\Gamma) = \underbrace{\begin{bmatrix} \mathbf{d}_1 \\ \vdots \\ \mathbf{d}_{m_d} \end{bmatrix}}_{\mathbf{d}}^T \underbrace{\begin{bmatrix} C(T_1) & & \\ & \ddots & \\ & & C(T_{m_d}) \end{bmatrix}^{-T}}_{C^{-T}} \underbrace{\begin{bmatrix} Q(T_1) & & \\ & \ddots & \\ & & Q(T_{m_d}) \end{bmatrix}}_{Q}$$

$$\begin{bmatrix} C(T_1) & & \\ & \ddots & \\ & & C(T_{m_d}) \end{bmatrix}^{-1} \begin{bmatrix} \mathbf{d}_1 \\ \vdots \\ \mathbf{d}_{m_d} \end{bmatrix} \quad (21)$$

$$= \begin{bmatrix} \mathbf{d_f} \\ \mathbf{d}_p \end{bmatrix}^T \underbrace{SC^{-T}QC^{-1}S^T}_{R} \begin{bmatrix} \mathbf{d}_f \\ \mathbf{d}_p \end{bmatrix} = \begin{bmatrix} \mathbf{d}_f \\ \mathbf{d}_p \end{bmatrix}^T \begin{bmatrix} R_{ff} & R_{fp} \\ R_{pf} & R_{pp} \end{bmatrix} \begin{bmatrix} \mathbf{d}_f \\ \mathbf{d}_p \end{bmatrix},$$

where $\mathbf{d}$ contains fixed derivatives ($\mathbf{d}_f$) and free derivatives ($\mathbf{d}_p$), and $S$ is a permutation matrix (ones and zeros), which is used to correct the order. Then, $\frac{dJ(\Gamma)}{d\mathbf{d}_p} = 0$ yields the optimal value for $\mathbf{d}_p$:

$$\mathbf{d}_p^* = -R_{pp}^{-1} R_{fp}^T \mathbf{d}_f. \tag{22}$$

Once $\mathbf{d}_p$ is determined, a polynomial that corresponds to each segment can be recovered.

### 4.5. Unconstrained Polynomial Trajectory Generation with Collision Avoidance

Oleynikova et al. [37] extended what Richter [35] proposed for adding support for collision avoidance capabilities. They added an additional term for calculating the collision cost:

$$J(\Gamma) = \xi_{obs} J_{obs}(\Gamma) + \xi_{smooth} J_{smooth}(\Gamma),$$
$$J_{smooth} = \mathbf{d}_f^T R_{ff} + \mathbf{d}_f^T R_{fp} \mathbf{d}_p + \mathbf{d}_p R_{pf} \mathbf{d}_f + \mathbf{d}_p^T R_{pp} \mathbf{d}_p, \tag{23}$$

where $J_{smooth}$ exactly equals (21). To estimate $J_{obs}(\Gamma)$, it is required to initially calculate position $\mathbf{p}_i(t)$ (16) and velocity $\mathbf{v}_i(t)$ for each axis at time $t$ after selecting the corresponding segment ($i, i = 1, \ldots, m_d$):

$$\mathbf{p}_i(t) = T p_i, \quad p_i = [\lambda_0, \lambda_1, \ldots, \lambda_d]_i^T, \quad T = [t^0, t^1, t^2, \ldots, t^d],$$
$$\mathbf{v}_i(t) = \dot{\mathbf{p}}_i(t) = TV p_i, \tag{24}$$
$$\mathbf{p}_i(t) = [\mathbf{p}_x(t) \ \mathbf{p}_y(t) \ \mathbf{p}_z(t)]_i, \quad \mathbf{v}_i(t) = [\mathbf{v}_x(t) \ \mathbf{v}_y(t) \ \mathbf{v}_z(t)]_i.$$

Knowing (the values of) $\mathbf{p}_i(t)$ and $\mathbf{v}_i(t)$, $J_{obs}(\Gamma_i)$ can be fully determined by:

$$J_{obs}(\Gamma_i) = \int_S c(\mathbf{p}_i(t)) ds = \int_{t=0}^{t^d} c(\mathbf{p}_i(t)) \|\mathbf{v}_i(t)\| dt = \sum_{t=0}^{t^d} c(\mathbf{p}_i(t)) \|\mathbf{v}_i(t)\| \Delta t$$

$$\frac{\partial J_{obs}(\Gamma_i)}{\partial d\mathbf{p}_i(t)} = \sum_{t=0}^{t^d} \|\mathbf{v}_i(t)\| \bigtriangledown_i c(T(C^{-1}S)_{pp}) \Delta t + c(\mathbf{p}_i(t)) \frac{\mathbf{v}_i(t)}{\|\mathbf{v}_i(t)\|} TV(C^{-1}S)_{pp} \Delta t, \tag{25}$$

where $(C^{-1}S)_{pp}$ is the right-side matrix, which corresponds to $\mathbf{d}_p$. For representing the collision cost $c(\mathbf{p}_i(t))$, a line integral of a potential function, i.e., (44), was used. As total cost is given (21), $J_{obs}(\Gamma)$ can be calculated for all the segments provided that $\mathbf{d}_p^*$ can be estimated. In a cluttered environment, optimization problem is most likely to be nonlinear as well as nonconvex. Thus, Broyden–Fletcher–Goldfarb–Shanno (BFGS) [75] was used to solve the optimization problem. However, the solver failed to obtain the global minimum most of the time. Hence, several random restarts were needed to find the optimal solution. A thorough discussion of how random restarts were invoked into the optimization problem was detailed in [45].

### 4.6. Covariant Gradients for Trajectory Generation

The significance of covariant gradients technique is that both $J_{obs}(\Gamma)$ and $J_{smooth}(\Gamma)$ depend solely on physical characteristic of the desired trajectory. In other words, the trajectory generation is invariant to its parameterization. If gradient descent is applied, it depends on the way trajectory is parameterized. The covariant gradients technique removes this dependency. Hence, covariant gradient technique depends solely on physical representation or dynamic quantities of the trajectory with respect to an operator, $\Theta$:

$$\|\Gamma\|_\Theta^2 = \int \sum_{n=1}^{k} \xi(\Gamma(t)^{(n)})^2 dt, \tag{26}$$

where $\xi$ is a constant and apices $^{(n)}$ determine the $n$th-order derivative. The correlation of derivatives between two trajectories: $\Gamma_1$ and $\Gamma_2$, which are defined by assuming inner product as given (27).

$$< \Gamma_1, \Gamma_2 >= \int \sum_{n=1}^{k} \xi \Gamma_1(t)^{(n)} \Gamma_2(t)^{(n)} dt. \tag{27}$$

The primary objective of $\Theta$ is to distinguish the norm (26) and the inner product (27) from the L2 norm [50].

*4.7. B-Spline-Based Trajectory Generation*

$d$th-order B-spline can be defined for a given knot sequence $p_k = \{t_0, t_1, \ldots, t_{n_k}\}$ and control points $p_c = \{\mathbf{p}_0, \mathbf{p}_1, \ldots, \mathbf{p}_{n_p}\}$, where $t_* \in \mathbb{R}$, $\mathbf{p}_* \in \mathbb{R}^d$ and $n_k = n_p + d + 1$. If $d$ is set to 3, each $\mathbf{p}_i$ represents position in $\mathbb{R}^3$, where $i = 0, \ldots, n_p$. For a given time index $t$, the corresponding position $\mathbf{p}(t)$ can be fully determined by using the de Boor–Cox formula [76].

$$\mathbf{p}(t) = DeBoorCox(t, p_c). \tag{28}$$

Estimation is not limited to the position; velocity, acceleration, or any high-order derivative of $p_c$ can be estimated using $DeBoorCox(t, p_c^{(*)})$, as given in Algorithm 1, where $(*)$ depicts the order of the derivative of $p_c$ such that $(*) < d$.

---

**Algorithm 1** The B-spline trajectory ($p$) and its derivative estimation for a given time index $t$, where $p$ equals $p_c^{(*)}$.

---

1: **procedure** DEBOORCOX($t, p$)

2: $\quad t = \begin{cases} p_k[d], & if\ t < p_k[d] \\ p_k[n_k], & if\ t > p_k[n_k] \\ t, & otherwise \end{cases}$

3: $\quad k = d$

4: $\quad$ **while** *true* **do**

5: $\quad\quad$ **if** $p_k[k+1] \geq t$ **then**

6: $\quad\quad\quad$ break

7: $\quad\quad$ $k$++

8: $\quad$ $\mathbf{p}_e[d]$

9: $\quad$ **for** $i \leftarrow 0 \quad to \quad d$ **do**

10: $\quad\quad$ $\mathbf{p}_e[i] \leftarrow p[k-d+i]$

11: $\quad$ **for** $r \leftarrow 1 \quad to \quad d$ **do**

12: $\quad\quad$ **for** $i \leftarrow d \quad to \quad r$ **do**

13: $\quad\quad\quad$ $\beta \leftarrow \frac{t - p_k[i+k-d]}{p_k[i+1+k-r] - p_k[i+k-d]}$

14: $\quad\quad\quad$ $\mathbf{p}_e[i] \leftarrow (1-\beta) \times \mathbf{p}_e[i-1] + \beta \times \mathbf{p}_e[i]$

15: $\quad$ **return** $\mathbf{p}_e[d]$

---

Later, the B-spline matrix representation was proposed by Qin [77]. B-spline can be formulated as uniform or nonuniform. J. Hu et al. [78] detailed the uniform B-spline matrix representation. In uniform B-spline, knot span is the same for any considered consecutive time interval, i.e., $\Delta t = t_{i+1} - t_i$, $i \in [0, n_k)$. Any position of the trajectory can be parameterized by considering only $d + 1$ consecutive control points: $[\mathbf{p}_i, \mathbf{p}_{i+1}, \ldots, \mathbf{p}_{i+d}]$. Hence, corresponding normalized time $q(t)$ can be calculated as follows:

$$q(t) = \frac{t - t_i}{t_{i+1} - t_i} = \frac{t - t_i}{\Delta t}, \quad t \in [t_i, t_{i+1}]. \tag{29}$$

In the matrix representation, $c(q(t))$, which is given in (28), can be determined by:

$$c(q(t)) = \mathbf{q}(t) M_d p_i, \quad \mathbf{q}(t) = [1, q(t), q^2(t), \ldots, q^d(t)]^T, \quad p_i = [\mathbf{p}_i, \mathbf{p}_{i+1}, \ldots, \mathbf{p}_{i+d}]^T,$$

$$M_d \in \mathbb{R}^{d+1 \times d+1}, \quad M_{r,c} = \frac{1}{d!} \binom{d}{d-r} \Sigma_{s=c}^{d} (-1)^{s-c} \times \binom{d}{s-c} (d-s)^{d+1-r-s}. \tag{30}$$

Since each control point $\mathbf{p}_i$ belongs to $d + 1$ of successive spans, B-spline can be controlled locally. Due to such controllability, B-spline is suitable for local trajectory planning [30]. Moreover, the derivatives of a given B-spline are also B-spline [8]. Hence, B-spline's derivatives (e.g., velocity, acceleration, jerk) can be calculated considering corresponding span $[t_i, t_i + 1)$ for a given $d + 1$ consecutive control points $p_i = [\mathbf{p}_i, \mathbf{p}_{i+1}, \ldots, \mathbf{p}_{i+d}]^T \in \mathbb{R}^{d \times 3}$ and corresponding knot vector.

$$\frac{dc(q(t))}{du} = \frac{1}{(\Delta t)} b_1 M_d \mathbf{v_i}^T, \quad b_1 = [0, 1, u, \ldots, u^{d-1}] \in \mathbb{R}^{d+1},$$

$$\frac{d^2 c(q(t))}{d^2 u} = \frac{1}{(\Delta t^2)} b_2 M_d \mathbf{v_i}^T, \quad b_2 = [0, 0, 1, u, \ldots, u^{d-2}] \in \mathbb{R}^{d+1}. \tag{31}$$

The explicit form of estimation of velocity and acceleration of a given time index is calculated as follows:

$$\frac{dc(q(t))}{du} = d \cdot \frac{p_c(i+1) - p_c(i)}{p_k(i+d+1) - p_k(i+1)},$$

$$\frac{d^2 c(q(t))}{d^2 u} = \tag{32}$$

$$(d^2 - d) \cdot \left( \frac{p_c(i+2) - p_c(i+1)}{p_k(i+d+2) - p_k(i+2)} - \frac{p_c(i+1) - p_c(i)}{p_k(i+d+1) - p_k(i+1)} \right).$$

In most of the situations, initial control points are generated, as explained in Section 3. Such methods may or may not be smooth enough for initial trajectory generation. There are various ways to generate intermediate waypoints to improve the quality of the trajectory using B-splines. For example, the initial trajectory was constructed using cubic B-Spline in [62]. Such a capability is mainly due to B-spline's properties.

It is particularly continuity and convex-hall properties that make B-spline trajectory generation such a robust technique.

### 4.7.1. Convex Hull Property

Among the properties of the B-spline, the convex hull property is the most significant property due to its capabilities for checking the dynamical feasibility and the collision. How convex hull property is incorporated for calculating dynamical feasibility is given in (32). As shown in Figure 3, $d_h > 0$ and $d_h > d_c - r_h$ should be held for a considered point in the trajectory to ensure a collision-free trajectory, where $d_c$ is the distance between a given control point and its closest obstacle position. In $d$th-order B-spline, a convex hull is formed by connecting any successive $d + 1$ control points, e.g., $\mathbf{p}_i, \mathbf{p}_{i+1}, \mathbf{p}_{i+2}, \ldots, \mathbf{p}_{i+d}$ or union of all consecutive control points that lie on the corresponding B-spline curve [69]. Moreover, $r_h$ can be substituted with $d_{i,i+1} + d_{i+1,i+2} + d_{i+2,i+3}$ since $r_h \leq d_{i,i+1} + d_{i+1,i+2} + d_{i+2,i+3}$, $d_h > d_c - (d_{i,i+1} + d_{i+1,i+2} + d_{i+2,i+3})$, where $d_{i,i+1} = \|\mathbf{p}_{i+1} - \mathbf{p}_i\|$, $d_{i+1,i+2} = \|\mathbf{p}_{i+2} - \mathbf{p}_{i+1}\|$ and $d_{i+2,i+3} = \|\mathbf{p}_{i+4} - \mathbf{p}_{i+3}\|$. As mentioned in [19], the following condition should hold for collision-free trajectory planning:

$$d_{i,i+1} < \frac{d_c}{3}, \quad d_c > 0, \quad i \in \{1, 2, 3\}. \tag{33}$$

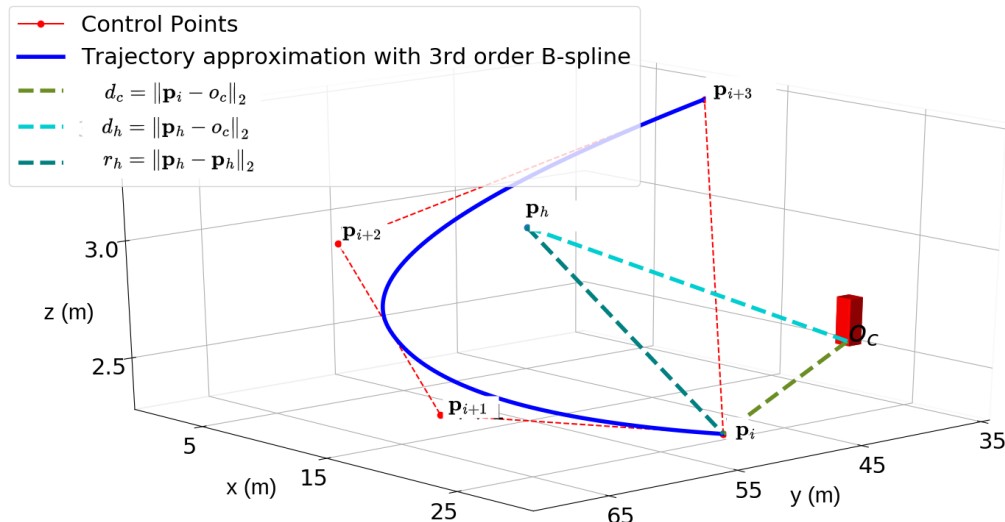

**Figure 3.** Showing the B-spline convex-hull property. Convex hull, which comprises consecutive control points, e.g., $\mathbf{p}_i, \mathbf{p}_{i+1}, \mathbf{p}_{i+2}$ and $\mathbf{p}_{i+3}$, always belongs to obstacle-free space if the preceding control points satisfy the inequality (33).

4.7.2. Continuity

B-spline-based trajectory generation offers several advantages over piece-wise-based trajectory generation [35,37]. In piece-wise-based trajectory generation, the boundary constraints must be explicitly satisfied to ensure continuity. The smoothness of the trajectory depends solely on how the control points are formed. In contrast, B-spline-based trajectory generation can neglect boundary constraints because the entire trajectory can be treated as one segment. Moreover, as explained in Section 4.7.1, B-spline-based trajectories can be controlled locally without affecting the rest of the trajectory.

*4.8. Bernstein Piece-Wise Trajectory Generation*

Bernstein polynomial is a specific form of B-spline, which is similar to the Bezier curve [79,80]. Bernstein polynomial can be described as follows:

$$\Gamma_j(t) = \lambda_j^0 p_d^0(t) + \lambda_j^1 p_d^1(t) + \cdots + \lambda_j^d p_d^d(t) = \Sigma_{i=0}^d \lambda_j^i p_d^i(t),$$

$$p_d^i(t) = \binom{d}{i} \cdot t^i \cdot (1-t)^{d-i}, \tag{34}$$

where $d$ is the degree of the polynomial (Figure 4), $\lambda_j^0, \lambda_j^1, \ldots, \lambda_j^d$ are the control points of $j$th polynomial segment, and $t \in [0,1]$. Since Bezier is a particular form of the B-spline curve, such curves hold convex hull property. Hence, given a sequence of control points, a constrained convex hull can be defined using the control points that are considered. Both the beginning and end of the curve are determined by the first and the last control points, respectively. Furthermore, the derivative of Bezier is also a Bezier curve.

$$\Gamma_\mu(t) = \begin{cases} s_1 \cdot \Sigma_{i=0}^d \lambda_{1,\mu}^i p_d^i(\frac{t-t_0}{s_1}) & t_0 \le t < t_1 \\ s_2 \cdot \Sigma_{i=0}^d \lambda_{2,\mu}^i p_d^i(\frac{t-t_1}{s_2}) & t_1 \le t < t_2 \\ \quad\vdots & \\ s_m \cdot \Sigma_{i=0}^d \lambda_{m_d,\mu}^i p_d^i(\frac{t-t_{m_d-1}}{s_{m_d}}) & t_{m_d-1} \le t < t_{m_d} \end{cases}, \tag{35}$$

where $i, j$ refer to $i$th control point in $j$th segment, i.e., $\lambda_j^i, s_j$ is a scaling factor of $j$th segment for mapping time duration from $[0,1]$ to $[t_{j-1}, t_j]$ and $\mu \in \{x, y, z\}$. Once $\Gamma_\mu(t)$ is obtained, the objective is to minimize the total cost, which can be determined by taking the integral

of square error up to $k_r$ order as given in (15). Such a problem can be formulated as a QP constraint problem. For instance, Gao and Wu [26] proposed a Bernstein-based trajectory optimization approach in which three types of constraints piece-wise trajectory continuity, safety constraints which are based on convex hull property, and dynamical feasibility constraints enforced [26].

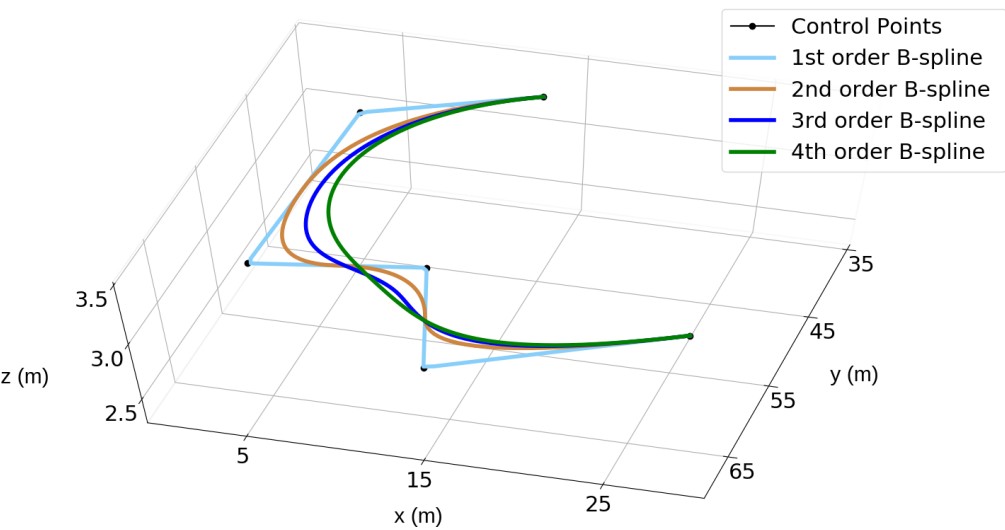

**Figure 4.** Trajectory generation using uniform B-spline. The smoothness of the curve is dependent on the degree of the B-spline. The trajectory passes precisely through the given control points at the degree equal to 1, as depicted in light blue color. The smoothness of the trajectory increases with the order of the B-spline.

### 4.9. Comparison of Several Trajectory Techniques

In the preceding subsections, several types of trajectory parameterization techniques were considered. We have selected three different types of trajectory parameterization techniques for this comparison: piece-wise-polynomials technique, fitting based on a sequence of points, and the third is uniform B-spline-based technique. The objective of piece-wise-polynomials is to find optimal polynomial coefficients [53] or end-derivatives [35] of consecutive segments, whereas the objective of the third technique is to find a set of points satisfying the provided constraints [57]. A comparison of how velocity, acceleration, jerk, and snap are varied for selected techniques in terms of mean, standard deviation (std), min, and max for the same a set of control points and knot vector is present below. The considered knot vector and control points are:

$$
\begin{aligned}
p_{ctrl} = [[0.011, -0.0329, 2.017], [1.867, 3.408, 1.6], [7.514, 5.715, 3.735], \\
[8.410, 0.911, 1.600], [6.902, -5.531, 4.306], [1.899, -6.680, 3.082], \\
[-2.302, -0.611, 5.375]] \\
p_{knot} = [0.0, 5.0, 12.0, 18.0, 26.0, 31.0, 40.0].
\end{aligned}
\tag{36}
$$

Each approach has its own set of parameters to fine-tune for obtaining an optimal trajectory. The generated trajectories are shown in Figure 5 with different configuration setups (with different parameter sets). Figure 6 shows how the derivatives up to the fourth change over time in each direction, i.e., x, y, z, separately for each technique. When looking at the derivatives of each method, it is clear that smoothness, which is the main point to be considered for motion planning, is higher in both B-spline and minimum-snap compared to CHOMP. Since uniform B-spline is used in this comparison, smoothness changes of each derivative between B-spline and minimum-snap cannot be compared directly due to time allocation when generating the trajectories. Hence, minimum-snap trajectory smoothness

can be changed, optimizing the time allocation process [35]. On the contrary, such a time allocation process is not necessary for a uniform B-spline. However, control points are interpolated appropriately to generate a continuous and smooth trajectory.

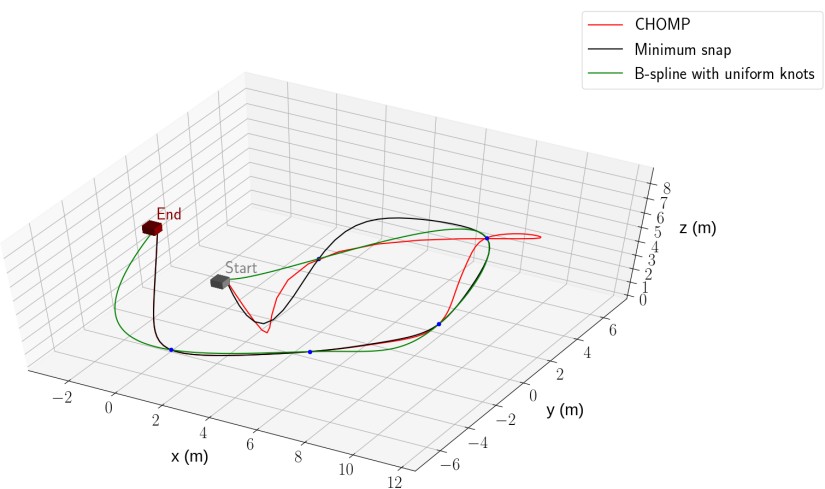

**Figure 5.** Generated trajectories using three different approaches for a given sequence of control points and knot vector (36).

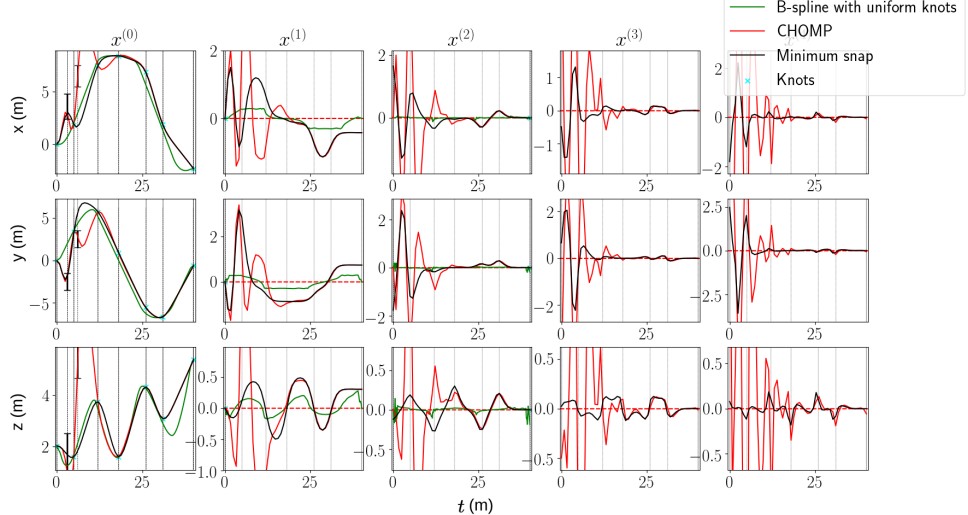

**Figure 6.** Changes of position, velocity, acceleration, jerk, and snap profiles over time for the provided control points sequence and knot vector (36).

We varied the parameters for each approach appropriately and estimated the mean, standard deviation, maximum, and minimum of the velocity, acceleration, jerk, and snap profiles. The results are shown in Table 1. The results clearly show that the consistency of the trajectory depends on the parameters used to parameterize the trajectory. Therefore, selecting the appropriate parameter set for a given task is of utmost importance, as can be seen by looking at the statistical properties (mean, standard deviation, minimum, and maximum) of the higher-order derivatives, such as velocity, acceleration, jerk, and snap. As described in the previous paragraph, the time allocation process directly affects the parameter selection for minimum-snap. Further, the optimal polynomial coefficients process depends on time allocation, as given in (13). On the other hand, the Poly-traj [35] generation process has fewer parameters to be optimized, since it uses free-end derivatives of each segment. Hence, the latter technique is faster than minimum-snap.

**Table 1.** Velocity, acceleration, jerk, and snap profile for generating an optimal trajectory for a given set of knot vector and control points (Figure 6) using three different techniques: minimum-snap [53], Poly-traj [35], and CHOMP [57].

| Type | Velocity | | | | Acceleration | | | |
|---|---|---|---|---|---|---|---|---|
| | Mean | Std | Min | Max | Mean | Std | Min | Max |
| Poly-traj, d: 8, mc: 2 | 0.0058 | 1.0154 | −1.4545 | 3.9179 | 0.0056 | 0.9051 | −2.835 | 3.6449 |
| Poly-traj, d: 8, mc: 6 | 0.0 | 0.0 | 0.0 | 0.0 | 0.0 | 0.0 | 0.0 | 0.0 |
| Poly-traj, d: 6, mc: 4 | 0.006 | 1.0708 | −1.7716 | 3.7864 | 0.0043 | 0.9307 | −2.7987 | 3.6032 |
| Poly-traj, d: 8, mc: 4 | 0.0059 | 1.0299 | −1.4728 | 3.934 | 0.0053 | 0.9131 | −2.9157 | 3.5214 |
| Poly-traj, d: 10, mc: 4 | 0.0058 | 1.0057 | −1.4428 | 3.9213 | 0.0052 | 0.8918 | −2.7541 | 3.631 |
| Minimum-snap, d: 8, mc: 2 | 0.1258 | 1.2154 | −1.4345 | 3.1259 | 0.0676 | 0.1259 | −2.2874 | 3.3278 |
| Minimum-snap, d: 8, mc: 6 | 0.0045 | 0.0094 | −0.07 | 0.019 | 0.09 | 0.0097 | −0.0098 | 0.0014 |
| Minimum-snap, d: 6, mc: 4 | 0.0689 | 1.0009 | −1.3416 | 3.2388 | 0.0012 | 0.4584 | −2.3189 | 3.2185 |
| Minimum-snap, d: 8, mc: 4 | 0.0015 | 1.0412 | −1.3215 | 3.7543 | 0.0075 | 0.8763 | −2.5487 | 3.3215 |
| Minimum-snap, d: 10, mc: 4 | 0.0036 | 1.0006 | −1.3428 | 3.7832 | 0.0099 | 0.4378 | −2.4548 | 3.4893 |
| CHOMP, pd: 3 | 0.0068 | 0.6421 | −0.9522 | 1.7255 | 0.0045 | 0.3876 | −1.131 | 1.476 |
| CHOMP, pd: 5 | 0.0065 | 0.644 | −0.9634 | 1.7161 | 0.0044 | 0.3909 | −1.1082 | 1.4418 |
| CHOMP, pd: 7 | 0.0064 | 0.6443 | −0.966 | 1.7105 | 0.0043 | 0.3916 | −1.0951 | 1.4205 |
| Type | Jerk | | | | Snap | | | |
| | mean | std | min | max | mean | std | min | max |
| Poly-traj, d: 8, mc: 2 | 0.007 | 1.2544 | −4.8056 | 3.9318 | −0.0151 | 2.3178 | −9.8029 | 6.9483 |
| Poly-traj, d: 8, mc: 6 | 0.0 | 0.0 | 0.0 | 0.0 | 0.0 | 0.0 | 0.0 | 0.0 |
| Poly-traj, d: 6, mc: 4 | 0.0117 | 1.568 | −5.5746 | 5.7423 | −0.1288 | 3.5271 | −13.4562 | 10.2578 |
| Poly-traj, d: 8, mc: 4 | −0.0021 | 1.2562 | −4.7192 | 3.7562 | 0.0131 | 1.9593 | −7.7131 | 6.0049 |
| Poly-traj, d: 10, mc: 4 | 0.0074 | 1.3399 | −5.5769 | 4.409 | −0.0504 | 3.1073 | −12.3429 | 9.9933 |
| Minimum-snap, d: 8, mc: 2 | 0.0006 | 1.1125 | −4.3413 | 3.5153 | −0.0042 | 2.1383 | −9.0056 | 6.3418 |
| Minimum-snap, d: 8, mc: 6 | 0.0005 | 0.0004 | −0.0007 | 0.0089 | 0.0005 | 0.004 | −0.0008 | 0.0009 |
| Minimum-snap, d: 6, mc: 4 | 0.01 | 1.3456 | −5.2167 | 5.321 | −0.0093 | 3.214 | −12.5124 | 9.2134 |
| Minimum-snap, d: 8, mc: 4 | −0.001 | 1.1321 | −3.7192 | 3.3217 | 0.0093 | 1.2145 | −5.6527 | 4.7854 |
| Minimum-snap, d: 10, mc: 4 | 0.0009 | 1.2145 | −3.9987 | 3.9983 | −0.0067 | 2.8731 | −10.7653 | 8.8416 |
| CHOMP, pd: 3 | 0.0021 | 0.3643 | −1.2594 | 1.1584 | −0.0014 | 0.4239 | −1.8326 | 1.5425 |
| CHOMP, pd: 5 | 0.0023 | 0.3628 | −1.2553 | 1.1639 | 0.0005 | 0.4241 | −1.8526 | 1.6243 |
| CHOMP, pd: 7 | 0.0022 | 0.3614 | −1.2732 | 1.1769 | 0.0021 | 0.4247 | −1.7462 | 1.5906 |

d: order of the polynomial, mc: maximum continuity or maximum continuity order in between consecutive segments, pd: number of proposed points or point density per defined time duration of the trajectory.

## 5. Free Space Extraction

Obstacle region identification is of utmost importance for optimal trajectory planning in real time. In a cluttered environment, the way the trajectory planning problem formulated matters for fast reaction. Such trajectory planning approaches can be designed as QP mainly due to less computation power required for such tasks. Hence, forming obstacle-free regions in the form of convex has more advantages in terms of reducing the computation power, simplicity, and fast convergence. Chen [74] attempted to define free space as a series of cubes between the start and goal pose. Thenceforth, OctoMap [81] was used for constructing the map surrounding the quadrotor, where regions with no obstacles are considered free spaces. After obtaining the free space information, obstacle constraints are enforced into (15) to generate optimal trajectory.

Let $C = [c_1^m, c_2^m, \ldots]$ be a set of consecutive grids within the OctoMap and corresponding free space regions be $C_{free} = [c_1^f, c_2^f, \ldots]$. Both $c_i^m$ and $c_i^f$ were defined as cubes, each of which is described by:

$$c_i^m = [\underbrace{c_{i_{x_0}}^m, c_{i_{y_0}}^m, c_{i_{z_0}}^m}_{l_m^i}, \underbrace{c_{i_{x_1}}^m, c_{i_{y_1}}^m, c_{i_{z_1}}^m}_{u_m^i}], \quad c_i^f = [\underbrace{c_{i_{x_0}}^f, c_{i_{y_0}}^f, c_{i_{z_0}}^f}_{l_f^i}, \underbrace{c_{i_{x_1}}^f, c_{i_{y_1}}^f, c_{i_{z_1}}^f}_{u_f^i}]. \tag{37}$$

Once $C_{free}$ was obtained, free space regions can be considered as a set of inequality constraints that can be added into the piece-wise-polynomials trajectory generation as $l_f^i \leq \Gamma_T(t_i) \leq u_f^i$, where $i = 1, \ldots, m_d - 1$ and $\Gamma_T$ was defined in (13). In such a trajectory, additional boundary constraints should be introduced if the extrema of $d$th-order polynomial violates the boundary constraints corresponding to each axis, i.e., x, y, and z, in each segment [74] (Equation (10)). Similar to the preceding approach, Gao and Shen [82] proposed a sequence of spheres to represent free space from the initial position to the final position. To construct the environment, a map was not built; instead, they bypassed map building by constructing a KD-tree-based placeholder [83] to store raw point cloud for the LiDAR. Afterwards, a relative map to the current pose of the MAV was retrieved using nearest neighbour search; RRG [84] combined with A* was used to find a flight corridor or intermediate waypoints. Such intermediate waypoints were connected by overlapping spheres.

IRIS [40] is one of the first successful ideas in which obstacle-free spaces are extracted using a convex optimization technique. In this proposed approach, initially, it is required to provide a seeking point and an area with a boundary box where an obstacle-free region is to be searched. Seeking point is defined as a unit ball: $\varepsilon(C, \mathbf{p}_0) = \{\mathbf{p} = C\tilde{\mathbf{p}} + \mathbf{p}_0 \mid \|\tilde{\mathbf{p}}\|_2 \leq 1\}$, where $\mathbf{p}_0$ is the center point. The linear constraints, which separate the boundary box into obstacle-free and obstacle-rich regions, are defined as a set of hyper-planes: $P = \{\mathbf{p} \mid A\mathbf{p} \leq b\}$. Subsequently, finding the optimal representation of $\varepsilon(C, \mathbf{p}_0)$ and $P$ with respect to given obstacles, $\imath_j, j = 1, \ldots, N$ is solved as an iterative process (38):

$$\begin{aligned}
\min_{C, \mathbf{p}_0, A, \mathbf{b}} \quad & -log(detC) \\
\text{s.t.} \quad & A_j^T \mathbf{p}_k \geq \mathbf{b}_j \quad \forall \mathbf{p}_k \in \imath_j, \quad j = 1, \ldots, N \\
& \sup_{\|\tilde{\mathbf{p}}\|} A_i^T(C\tilde{\mathbf{p}} + \mathbf{p}_0) \leq \mathbf{b}_i \quad \forall i = 1, \ldots, N,
\end{aligned} \tag{38}$$

where $A_i$ and $\mathbf{b}_i$ correspond to $i$th row of $A$ and $\mathbf{b}$. The first constraint, i.e., $A_j^T \mathbf{p}_k \geq \mathbf{b}_j$, is imposed to move the obstacle into one side of the plane, $A_j^T \mathbf{p} = \mathbf{b}_j$, whereas the second constraint, i.e., $\sup_{\|\tilde{\mathbf{p}}\|} A_i^T(C\tilde{\mathbf{p}} + \mathbf{p}_0) \leq \mathbf{b}_i$, ensures the ellipsoid lies on the other side of the plane. The researchers proposed to solve the (38) as a two-step process: searching, first, for proper constraints (i.e., $A_i$ and $\mathbf{b}_i$), and then the maximum volume that satisfies ellipsoid, ensuring preceding constraints. In other words, they attempted to find hyperplanes that separate obstacle regions and free regions. Conceptually, hyperplane separation is completed by finding planes that intersect with obstacle boundaries. Afterwards, the ellipsoid is uniformly expanded until it intersects with obstacle boundaries. Let $\alpha$ be the scaling factor which defines the expansion. Let $\varepsilon_\alpha = \{C\tilde{\mathbf{p}} + \mathbf{p}_0 \mid \|\tilde{\mathbf{p}}\|_2 \leq \alpha\}$ for $\alpha \geq 1$ be the expanded ellipsoid. Hence, the optimal $\alpha^*$ can be determined by:

$$\begin{aligned}
\alpha^* = \arg\min_{\alpha} \quad & \\
\text{s.t.} \quad & \varepsilon_\alpha \cap \imath_j \neq \varnothing.
\end{aligned} \tag{39}$$

After finding $\alpha^*$, it is possible to define the optimal inscribed ellipsoid ($\varepsilon^*$), which is the obstacle-free region [40] (Section 3.3).

Sikang et al. [32] proposed a new approach, quite different from the aforementioned IRIS, for extracting obstacle-free regions as a convex set SFC (Figure 7). SFC searches a set of overlapping polyhedra from the start pose to the goal pose. To obtain intermediate obstacle-free positions, the researchers utilized a graph search technique, namely JPS [55]. The main reason for selecting JPS over sampling-based algorithms (e.g., RRT* and PRM) or search-based techniques such as A* or Dijkstra is due to the nature of JPS; it uses a uniform-cost grid map with uniform voxels. In general, sampling-based techniques are not deterministic though probabilistically complete. Thus, there is no guarantee about the duration of searching time. On the other hand, the computational time for search-based methods is pretty high if the environment is cluttered. However, JPS has a lower searching time compared to A* [32]. Let $p_c = \mathbf{p}_0, \mathbf{p}_1, \dots, \mathbf{p}_n$ be the intermediate waypoints from start to goal pose and $l_i = < \mathbf{p}_i, \mathbf{p}_{i+1} >$ be the $i$th line segment, where $i = 1, \dots, n - 1$. Each line segment constitutes convex polyhedra, namely, $E_i$. Along with that, SFC can be expressed as $SFC(P) = \{ E_i \mid i = 0, \dots, n - 1 \}$. SFC has two steps: finding $E_i$ that fits the $l_i$ and seeking a set of linear inequalities that are tangent to $E_i$. Let $E_i$ be $\varepsilon_i(C_i, \mathbf{p}_i^0) = \{ \mathbf{p} = C_i \tilde{\mathbf{p}} + \mathbf{p}_i^0 \mid \| \tilde{\mathbf{p}} \|_2 \leq 1 \}$. In $\mathbb{R}^3$, $C_i$ can be decomposed as $R^T S R$, where $R$ gives the axis of rotation between considered line segment in between $\mathbf{p}_i$ and $\mathbf{p}_{i+1}$). The semi-axis of $E_i$ is given by $S = diag(a, b, c)$ as a diagonal matrix. $\mathbf{p}_i^0$ is the center of $l_i$. The objective of SFC is to find each pair $E_i$ and $\mathbf{p}_0^i$, given the $l_i$ and obstacles set ($Obs_i$), which touches the $E_i$.

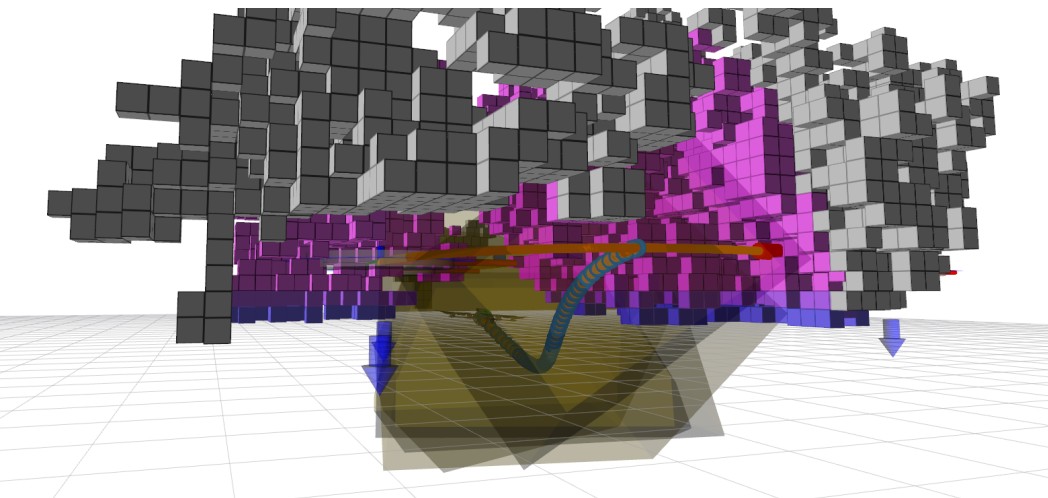

**Figure 7.** Free space extraction using SFC. Once intermediate initial waypoints are defined, SFC calculates free space along the path, which is constructed from the initial waypoints.

Initially, ellipsoids are spheres whose center poses are located as the midpoints of $l_i$, $i = 1, \dots, n - 1$. Afterwards, semi-axes, except for the axis along $\mathbf{p}_{i+1} - \mathbf{p}_i$, are shrunk until the corresponding ellipsoid contains no obstacles. Let $\varepsilon_i^*(C_i, \mathbf{p}_i^0)$ be the $i$th ellipsoid after applying the shrinking process. $\mathbf{p}_j$ denotes the closest point that touches the $\varepsilon_i^*(C_i, \mathbf{p}_i^0)$, where $j = 1, \dots, m$ and $m$ is the number of obstacles. Hence, corresponding half-space $H_j = \{ \mathbf{p}_j \mid a_j^T \mathbf{p}_j < \mathbf{b}_j \}$ is defined as a plane that is tangential to $\varepsilon_i^*(C, \mathbf{p}_0)$, where $a_j$ and $\mathbf{b}_j$ are determined by:

$$\mathbf{a}_j = \frac{d\varepsilon_i}{dp} \bigg|_{\mathbf{p}=\mathbf{p}_j} = 2C_i^{-1}C_i^{-T}(\mathbf{p}_j - \mathbf{p}_i^0), \quad \mathbf{b}_j = \mathbf{a}_j^T \mathbf{p}_j. \tag{40}$$

Hence, the intersections of these m half spaces create a convex polyhedron $C = \cup_{j=0}^m H_j = \{ \mathbf{p} \mid A^T \mathbf{p} < \mathbf{b} \}$. The same approach is applied to each line segment, $l_i$ in which we can generate each $C_i$. All in all, $SFC(P) = \{ C \mid i = 0, \dots, n - 1 \}$ can be constructed. A more descriptive formulation is given in [32] (Algorithm 1).

## 6. Continuous Trajectory Refinement

The objective function consists of several subobjective functions: for improving the smoothness, for avoiding obstacles, and so forth. In this section, a precise explanation is given on how to construct subobjective functions for each of the various occasions. First, we examine the simplest case where only dynamic feasibility and obstacle avoidance constraints are taken into consideration. Let $J$ be the objective function or performance index:

$$J(\Gamma) = \xi_{smooth} J_{smooth}(\Gamma) + \xi_{obs} J_{obs}(\Gamma). \tag{41}$$

There are various formulations of how $J_{obs}$ and $J_{smooth}$ are determined. In general, $J_{smooth}$ can be expressed as:

$$J_{smooth}(\Gamma) = \frac{1}{2} \int_0^1 \left\| \frac{d\Gamma(t)}{dt} \right\|^2 dt. \tag{42}$$

Eliminating unnecessary higher-order motion is the main objective of the $J_{smooth}$. On the other hand, $J_{obs}$ encourages to generate or modify collision-free trajectory by trying to push control points away from the obstacle zone if the trajectory is already in collisions or penalizing parts of the trajectory that is close to the obstacles. Let $B \subset \mathbb{R}^d$ be the exterior boundary of the MAV and $c$ is the cost function of penalizing close-in obstacles with respect to $B$. Along with that, $J_{obs}$ can be formulated as follows:

$$J_{obs}(\Gamma) = \int_0^1 \int_{u \in B} c(f_c(\Gamma(t), \mathbf{p})) \left\| \frac{df_c(\Gamma(t), \mathbf{p})}{dt} \right\|^2 d\mathbf{p} dt, \tag{43}$$

where the function $f_c(\Gamma(t), \mathbf{p})$, which was proposed by Ratliff at al. [57], can be defined as follows:

$$f_c(\Gamma(t), \mathbf{p}) = \begin{cases} -dis(\Gamma(t), \mathbf{p}) + \frac{1}{2}\delta_{dis} & if\ dis(\Gamma(t), \mathbf{p}) < 0 \\ \frac{1}{2\delta_{dis}}(dis(\Gamma(t), \mathbf{p}) - \delta_{dis})^2 & if\ 0 < dis(\Gamma(t), \mathbf{p}) \leq \delta_{dis}, \\ 0 & otherwise \end{cases} \tag{44}$$

where $\delta_{dis}$ denotes the distance from the boundary ($B$) of the quadrotor to a given obstacle position. Before taking gradient at $i$, $J(\Gamma)$ is linearized around $i$, $J(\Gamma) \approx J(\Gamma_i) + (\Gamma - \Gamma_i)^T \bigtriangledown J(\Gamma_i)$. Defining $c$ and $d$ is detailed in [57] (Equations (22)–(28)).

In [19], the cost of the trajectory was estimated based on the following formulation:

$$J(\Gamma) = \xi_{obs} J_{obs}(\Gamma) + \xi_{smooth} J_{smooth}(\Gamma) + \xi_{soft} J_{soft}(\Gamma), \quad J_{soft}(\Gamma) = J_v(\Gamma) + J_a(\Gamma), \tag{45}$$

where $J_{soft}(\Gamma)$ is determined by soft limits on acceleration and velocity. $J_{smooth}(\Gamma)$ is defined by considering only geometric information without minimizing snap and/or jerk [53]. Such minimization is required because of the following stages of trajectory optimization. In such trajectory optimization, time reallocation has less impact on the objective function. Hence, $J_{smooth}(\Gamma)$ is defined as follows:

$$J_{smooth}(\Gamma) = \Sigma_{i=d-1}^{n+1-d} \left\| \underbrace{\mathbf{p}_{i+1} - \mathbf{p}_i}_{f_{i+1,i}} + \underbrace{\mathbf{p}_{i-1} - \mathbf{p}_i}_{f_{i-1,i}} \right\|^2, \tag{46}$$

where a number of control points, denoted n, and $\mathbf{f}_{i+1,i}$ and $\mathbf{f}_{i-1,i}$ can be interpreted as connecting joint force of two springs between control points pairs: $(\mathbf{p}_{i+1}, \mathbf{p}_i)$ and $(\mathbf{p}_{i-1}, \mathbf{p}_i)$, for example, control points lie on a straight line if the sum of all terms equals zero. As an aside, similar approaches were proposed in [85,86].

The value of $J_{obs}(\Gamma)$ is determined by calculating the distance to the closest object pose from each control point, in which the distance to the obstacle, i.e., $f_c(\mathbf{p}_i)$, is given by:

$$f_c(\mathbf{p}_i) = \begin{cases} (dis(\mathbf{p}_i) - \delta)^2 & dis(\mathbf{p}_i) \le \delta_{dis} \\ 0 & dis(\mathbf{p}_i) > \delta_{dis} \end{cases}, \tag{47}$$

where $\delta_{dis}$ is the free distance between MAV's center and the pose of the closest obstacle. Hence, $J_{obs}(\Gamma) = \Sigma_{i=d}^{n} f_c(\mathbf{p}_i)$ can be estimated based on a given trajectory in the form of control points. Soft constraints are defined by not exceeding both acceleration and velocity within those max limits.

$$J_{\mathbf{v}}(\Gamma) = \sum_{\mu} \sum_{i=d-1}^{n-d} f_v(\mathbf{v}_{i,\mu}), \quad J_a(\Gamma) = \sum_{\mu} \sum_{i=d-2}^{k_d-d} f_a(\mathbf{a}_{i,\mu}) \tag{48}$$

$$f(\mathbf{v}) = \begin{cases} (\mathbf{v}_\mu^2 - \mathbf{v}_{max}^2)^2 & \mathbf{v}_\mu^2 > \mathbf{v}_{max}^2 \\ 0 & \mathbf{v}_\mu^2 \le \mathbf{v}_{max}^2 \end{cases}, \quad f(\mathbf{a}) = \begin{cases} (\mathbf{a}_\mu^2 - \mathbf{a}_{max}^2)^2 & \mathbf{a}_\mu^2 > \mathbf{a}_{max}^2 \\ 0 & \mathbf{a}_\mu^2 \le \mathbf{a}_{max}^2 \end{cases}.$$

To calculate acceleration and velocity at each control point and when both acceleration and velocity exceed their maximum limits, the convex hull property (33) of B-spline is utilized to penalize those control points. Based on the previous method, Ref. [30] proposed an endpoint cost $J_{endpoint}(\Gamma)$ into the objective function as an additional term. The key intuition behind adding $J_{endpoint}(\Gamma)$ is to reduce the error between local trajectory and global trajectory since $J_{endpoint}(\Gamma)$ penalizes error of both velocity and position with respect to the desired global trajectory. $J_{endpoint}(\Gamma)$ is determined as follows:

$$J_{endpoint}(\Gamma) = J_{end}(\Gamma) = \xi_{end}^p(\mathbf{p}(t_{end}) - \mathbf{p}_{end})^2 + \xi_{end}^v(\dot{\mathbf{p}}(t_{end}) - \dot{\mathbf{p}}_{end})^2, \tag{49}$$

where $\xi_{end}^p$ and $\xi_{end}^v$ are regularization parameters, whereas $\mathbf{p}_{end}$ and $\dot{\mathbf{p}}_{end}$ are the desired end position and velocity of the trajectory.

## 7. Receding Horizon Trajectory Planning

On most occasions, paths which are obtained by planning techniques are suboptimal. Hence, the initial trajectory that is generated based on the initial path is to be refined, ensuring dynamic feasibility for controlling the MAV. Various approaches can be applied for trajectory refinement. However, enabling recursive feasibility and incorporating terminal constraints and convergence to the desired state are the utmost importance considerations to be contemplated throughout the process. LQR and MPC are the two most popular approaches that are being used for receding horizon planning. LQR is applied for linear systems, whereas iLQR and differential dynamic programming (DDP) are applied for nonlinear system. Both in LQR or iLQR, OCP is defined as an open-loop control problem. On the other hand, MPC is designed as a close-loop OCP. In other words, OCP is seeking actions knowing the behavior of the surrounding environment.

### 7.1. LQR-Based Trajectory Generation

DDP [87,88] is one of the first techniques proposed for solving optimal control problems. Let $\mathbf{x}_{k+1} = \mathbf{f}_d(\mathbf{x}_k, \mathbf{u}_k)$ be the discrete-time system dynamics; the total cost of the trajectory can be formulated for a given control policy, i.e., $\pi_{k+i}$, for all $i = \{0, 1, \ldots, N-1\}$.

$$\sum_{i=0}^{N-1} c(\mathbf{x}_{k+i}, \mathbf{u}_{k+i}) + c_{goal}(\mathbf{x}_{k+N}). \tag{50}$$

The optimal control input, i.e., $\mathbf{u}_{k+i} = \pi_{k+i}(\mathbf{x}_{k+i})$, for a given time index, i.e., $i + k$, can be obtained by minimizing the (50). Thus, cost (cost-to-go), which was proposed in [89], is fully determined by:

$$V_{k+i}(\mathbf{x}_{k+i}) = \min_{\mathbf{u}_{k+i}} (c(\mathbf{x}_{k+i}, \mathbf{u}_{k+i}) + V_{k+1}(\mathbf{f}_d(\mathbf{x}_{k+i}, \mathbf{u}_{k+i})). \tag{51}$$

The same procedure can be applied recursively in a backward direction for seeking the optimal $\pi_{k+i}(\mathbf{x}_{k+i}) = \arg\min_{\mathbf{u}_{k+i}}(c(\mathbf{x}_{k+i}, \mathbf{u}_{k+i}) + V_{k+i}(\mathbf{f}_d(\mathbf{x}_{k+i}, \mathbf{u}_{k+i})))$. DDP yields almost the same behavior: first estimate optimal control and then apply a forward pass to determine the updated nominal trajectory. Consequently, LQR is a simplified version

of DDP. LQR is one of the fundamental ways to obtain a closed-form solution for a given optimal control problem under which system dynamics is assumed to be linear. Let us assume the system dynamics is defined as in (4). The intuition of LQR is to estimate the optimal control sequence for maneuvering the quadrotor from an initial position to the desired pose. Let $N$ be the receding horizon whose optimal trajectory is to be determined. The total cost, i.e., $J_k(\mathbf{x}_k, \pi_N)$, consists of three parts: initial, intermediate, and final costs, where $\pi_N = \{\pi_k, \pi_{k+1}, \ldots, \pi_{k+i}, \ldots, \pi_{N-1}\}$:

$$J_k(\mathbf{x}_k, \pi_N) = c_{start}(\mathbf{x}_k) + \sum_{i=0}^{N-1} c(\mathbf{x}_{k+i}, \mathbf{u}_{k+i})dt + c_{end}(\mathbf{x}_{k+N}), \tag{52}$$

where $\frac{\partial^2 C_{start}(x_k)}{\partial x \partial x} \leq 0,\quad \frac{\partial^2 C_{goal}(x_{k+N})}{\partial x \partial x} \leq 0,\quad \frac{\partial^2 C}{\partial \begin{bmatrix} x \\ u \end{bmatrix} \partial \begin{bmatrix} x \\ u \end{bmatrix}} \leq 0$, and $\frac{\partial^2 C}{\partial u \partial u} \leq 0$ are positive semi-

definite Hessians to guarantee the minimizing of the total cost. The total cost can be formulated in various ways. In LQR, the total cost is defined as Quadratic costs as follows:

$$c_{start}(\mathbf{x}_k) = \frac{1}{2}\mathbf{x}_k^T Q_{start}\mathbf{x}_k + \mathbf{x}_k^T q_{start},$$

$$c_{goal}(\mathbf{x}_{k+N}) = \frac{1}{2}\mathbf{x}_{k+N}^T Q_{goal}\mathbf{x}_{k+N} + \mathbf{x}_{k+N}^T q_{goal},$$

$$c(\mathbf{x}_{k+i}, \mathbf{u}_{k+i}) = \frac{1}{2}\mathbf{x}_{k+i}^T Q\mathbf{x}_{k+i} + \frac{1}{2}\mathbf{u}_{k+i}^T R\mathbf{u}_{k+i} + \mathbf{u}_{k+i}^T P\mathbf{x}_{k+i} + \mathbf{x}_{k+i}^T p \tag{53}$$

$$+\mathbf{u}_{k+i}^T r + \xi = \frac{1}{2}\begin{bmatrix} \mathbf{x}_{k+i} \\ \mathbf{u}_{k+i} \end{bmatrix}^T \underbrace{\begin{bmatrix} Q & P^T \\ P & R \end{bmatrix} \begin{bmatrix} \mathbf{x}_{k+1} \\ \mathbf{u}_{k+1} \end{bmatrix}_{k+i}}_{J_k} + \begin{bmatrix} \mathbf{x}_{k+1} \\ \mathbf{u}_{k+1} \end{bmatrix}\underbrace{\begin{bmatrix} p \\ r \end{bmatrix}}_{j_k} + \xi,$$

where $i \in \{0, 1, \ldots, N-1\}$, $Q_{start} \in \mathbb{R}^{n_x \times n_x}, Q_{goal} \in \mathbb{R}^{n_x \times n_x}, Q \in \mathbb{R}^{n_x \times n_x}, R \in \mathbb{R}^{n_u \times n_u}$, $P \in \mathbb{R}^{n_u \times n_x}, q_{start} \in \mathbb{R}^{n_x}, q_{goal} \in \mathbb{R}^{n_x}, p \in \mathbb{R}^{n_x}, r \in \mathbb{R}^{n_u}$, and $\xi \in \mathbb{R}$ are predefined in which $Q_{start}, Q_{goal}, Q$, and $R$ are positive definite, whereas $J_k \geq 0$ and $j_k \geq 0$ assumed to be positive semi-definite. LQR problem (52) and (53) provides an optimal $\pi_N$ in close form solution as expressed in (51); the cost-to-go function, i.e., (51), can be reformulated as an explicit quadratic formulation as follows:

$$V_{k+i}(\mathbf{x}_{k+i}) = \frac{1}{2}\begin{bmatrix} \mathbf{x}_{k+i} \\ \mathbf{u}_{k+i} \end{bmatrix}^T J_{k+i} \begin{bmatrix} \mathbf{x}_{k+i} \\ \mathbf{u}_{k+i} \end{bmatrix} + \begin{bmatrix} \mathbf{x}_{k+i} \\ \mathbf{u}_{k+i} \end{bmatrix}^T j_{k+i} + \xi. \tag{54}$$

The estimation of both $J_{k+i}$ and $j_{k+i}$ can be obtained in a recursive way starting from the goal position $\mathbf{x}_{x+N}$ to the initial position $\mathbf{x}_k$, using Riccati differential equation for all $i = \{0, \ldots, N-1\}$.

$$J_k = Q + A_k^T J_{k+1} A_k -$$
$$(P + B_k^T J_{k+1} A_k)^T \cdot (R + B_k^T J_{k+1} B_k)^{-1} \cdot (P + B_k^T J_{k+1} A_k)$$
$$j_k = p + A_k^T j_{k+1} + A_k^T J_{k+1} c_k \tag{55}$$
$$-(P + B_k^T J_{k+1} A_k)^T \cdot (R + B_k^T J_{k+1} B_k)_k^{-1} \cdot (r + B_k^T j_{k+1} + B_k^T J_{k+1} c_k).$$

In general, system dynamics is described by:

$$\mathbf{x}_{k+1} = \mathbf{f}_d(\mathbf{x}_k, \mathbf{u}_k) = A_k\mathbf{x}_k + B_k\mathbf{u}_k. \tag{56}$$

If the system dynamics is nonlinear, $A_k$ and $B_k$ are recalculated by linearizing the $\mathbf{f}_c$ at each time index. Since linearization has to be carried out in each iteration, it is called the iLQR [90].

$$A_k = \frac{\partial \mathbf{f}_c}{\partial \mathbf{x}}(\mathbf{x}_k, \mathbf{u}_k), \quad B_k = \frac{\partial \mathbf{f}_c}{\partial \mathbf{u}}(\mathbf{x}_k, \mathbf{u}_k). \tag{57}$$

Boundary or goal position conditions are given by $S_{k+N} = Q_{goal}, \quad j_{k+N} = q_{goal}$. The feedback control policy in LQR is fully determined as follows:

$$
\begin{aligned}
\pi_k(\mathbf{x}_k) = &-(R + B_k^T J_{k+1} B_k)^{-1} \cdot (P + B_k^T J_{k+1} A_t) \mathbf{x}_k \\
&-(R + B_k^T J_{k+1} B_k)^{-1} \cdot (r + B_k^T j_{k+1} + B_k^T J_{k+1} B_k).
\end{aligned} \tag{58}
$$

As given in (55), system stability depends on system dynamics. When quadrotor dynamics is nonlinear, the stability of iLQR is not guaranteed. Jur and Berg [91] attempted to address the stability issue by proposing a novel method called LQR smoothing; this method can be applied for linear or nonlinear systems to acquire the minimum-cost trajectory. The main difference in LQR smoothing compared to LQR is that LQR minimizes the cost of not only backward direction, i.e., cost-to-go, but also applies forward direction, i.e., cost-to-come [58,91,92]. However, the output of LQR, iLQR or LQR smoothing does not address the system noise. Both linear or nonlinear state estimator may eliminate the system noise. LQG [93,94] is one of the ways to solve this problem. LQG consists of a state estimator, i.e., Kalman filter (KF), and state feedback, i.e., iLQR or LQR.

### 7.2. MPC-Based Trajectory Generation

As detailed in Section 7.1, unaccountability of addressing sudden disturbances is the main limitation of OCP techniques (e.g., LQR, DDP); this is due to its nature. LQR calculates fixed receding control policy and applies it to the system; there is no intervention during the control policy execution. MPC is one of the ways to address the preceding problem, which is characteristic of both LQR and DDP. The difference between MPC and LQR is that only the first portion of the control policy is applied to a system (Figure 8) in MPC through the calculation of full control policy rather than employing full control policy as in LQR. Let us assume the system dynamics as given in (2). In general, MPC can be formed as follows:

$$
\begin{aligned}
\min_{w} \quad & J_{end}(\mathbf{x}_{k+N}, \mathbf{x}_{k+N}^{ref}) + J_k(\mathbf{x}, \mathbf{u}, \mathbf{x}^{ref}, \mathbf{u}^{ref}) \\
\text{s.t.} \quad & \mathbf{x}_{k+1} = \mathbf{f}_d(\mathbf{x}_k, \mathbf{u}_k) \\
& \mathbf{x}^{min} \leq \mathbf{x}_{k+i} \leq \mathbf{x}^{max} \quad \forall 0 \leq i \leq N \\
& \mathbf{u}^{min} \leq \mathbf{u}_{k+i} \leq \mathbf{u}^{max} \quad \forall 0 \leq i < N-1 \\
& g_1(w) = 0 \\
& g_2(w) \leq 0,
\end{aligned} \tag{59}
$$

where $w = \mathbf{u}_k, \ldots, \mathbf{u}_{k+N-1}$ is the optimal control sequence to be estimated in each iteration. Variable $J_{end}(\mathbf{x}_{k+N})$ plays a significant role in terms of the stability of the system locally and globally. Presenting local stability is relatively easy, e.g., Lyapunov's analysis compared to global stability. In addition to terminal cost, terminal constraints for states should be enforced, which is quite computationally challenging for real-time applications. Moreover, enforcing terminal constraints is even more difficult for nonlinear dynamics. Thus, in most of the practical applications, terminal constraints are not enforced into the optimization procedure. Furthermore, classical MPC lacks recursive feasibility. Several varieties of MPC have been proposed to address processing issues to a certain extent. For a linear system, the performance index, i.e., $J_k(\mathbf{x}, \mathbf{u}, \mathbf{z}^{ref}, \mathbf{u}^{ref})$, can be defined as follows:

$$
J_k(\mathbf{x}, \mathbf{u}, \mathbf{z}^{ref}, \mathbf{u}^{ref})
$$

$$
= \sum_{i=0}^{N-1} \left( (\mathbf{x}_{k+i} - \mathbf{x}_{k+i}^{ref})^T Q_x (\mathbf{x}_{k+i} - \mathbf{x}_{k+i}^{ref}) + (\mathbf{u}_{k+i} - \mathbf{u}_{k+i}^{ref})^T R_u (\mathbf{u}_{k+i} - \mathbf{u}_{k+i}^{ref}) \right) \tag{60}
$$

$$
+ (\mathbf{x}_{k+N} - \mathbf{x}_{k+N}^{ref})^T P (\mathbf{x}_{k+N} - \mathbf{x}_{k+N}^{ref}),
$$

where $Q_x$, which is a positive semi-definite matrix, consists of the state error penalty coefficients, whereas $R_u$ should be positive definite and $P$ is state error on the terminal cost.

In principle, stability and feasibility are not assured implicitly. Consequently, stability and feasibility tend to improve for the longer receding horizon, which is quite challenging due to computational demands.

Quadrotor dynamics are usually expressed in a nonlinear fashion. Therefore, LQR or linear MPC cannot be applied without linear approximation. Hence, the nonlinear-programming-based approach has to be applied. Direct multiple shooting and direct collocation are the main two techniques that are used to transform OCP into nonlinear programming (NLP). In both direct multiple shooting and direct collocation, the state is minimized in addition to controlling inputs. Direct multiple shooting differs from direct collocation due to the way of the problem formulation. In multiple shooting, the problem is quantized into $N$ subintervals, i.e., receding horizon length. In direct collocation, however, those subintervals are further described by a set of polynomials such as B-spline or Lagrangian; this will increase the problem sparsity. On the contrary, the number of optimization parameters to be optimized has dramatically increased in direct collocation compared to multiple shooting. This, collocation is better when it is accuracy-wise, but direct multiple shooting is better when it is performance-wise. In [62], the trajectory tracking problem is formulated based on direct collocation and multiple shooting. Furthermore, the researchers have proven that multiple shooting has a lower computational footprint compared to direct collocation.

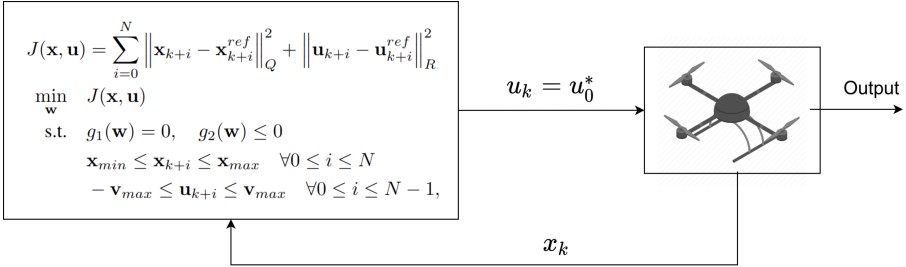

**Figure 8.** The basic idea of MPC-based receding horizon planning. MPC-based receding horizon planning predicts the optimal control policy $\mathbf{u}^*$ at each iteration to minimize the given cost function $J$.

### 7.3. Disturbance Estimation

In the context of optimal trajectory planning, simultaneously computing optimal control policy, which is required to respond to unknown, sudden disturbances, and handling kinematics (i.e., obstacle avoidance) as well as dynamics (i.e., satisfying velocity and acceleration constraints) yields a challenging problem, especially for quadrotors. While geometry-based path planning techniques [95,96] ensure the asymptotical optimality of a path, they however do not consider quadrotor dynamics. However, it is essential that the generation of an optimal control policy ensures dynamic feasibility. Thus, in [97,98], LQR was incorporated into path planning, by which both dynamic feasibility and local optimality were guaranteed. However, local optimality does not necessarily yield global optimality [99]. In [26,32], a set of motion primitives was used to find feasible trajectories ensuring both global and local optimality. When dealing with unknown disturbances, MPC is a more robust technique than LQR. In [32], MPC-based trajectory planning approach was proposed, ensuring both the local and global optimality. However, none of the aforementioned approaches formally guarantees stability and safety. Lyapunov's analysis can be applied to confirm the local stability. Moreover, the terminal constraints set [100] can be incorporated. However, those measures are time consuming, which directly affects the real-time performance [101]. A set of CBFs was proposed for improving real-time performance without affecting the system stability in [102–104]. Recently, reference governors-based techniques were proposed in [105,106], enforcing safety constraints. It is natural that designing a path planer is followed by the actual controller to maneuver MAV. In such approaches, a reference governor can be used to handle the stability and constraint satisfaction separately to ensure system stability [107].

The above approaches are employed to estimate optimal control policy for safe navigation while imposing stability either using Lyapunov functions or reference governors. On the other hand, Li et al. [108] proposed to obtain an optimal control policy using a state-dependent distance metric (SDDM). They have modeled the system dynamics as a linear, time-invariant as follows:

$$\dot{\mathbf{x}} = A\mathbf{x} + B\mathbf{u}, \tag{61}$$

where $\mathbf{u}$ indicates the control input. The system state, i.e., $\mathbf{x} := (\mathbf{p}(t), \mathbf{y}(t))$, consists of two parts: $\mathbf{p}$ and $\mathbf{y}$, where $\mathbf{p}(t)$ denotes the quadrotor position at a given time $t$ and $\mathbf{y}(t)$ describes the higher-order terms, e.g., velocity, acceleration, etc. In the latter work, the quadratic norm was utilized to represent the error between robot position and close-in obstacles positions. The quadratic norm is defined as $\|\mathbf{p}\|_R := \sqrt{\mathbf{p}^T R \mathbf{p}}$, where $R$ is a symmetric positive matrix. $R[\psi_z]$ is fully determined by the MAV heading direction $\psi_z$ at a given time instance as follows:

$$R[\psi_z] = \begin{cases} c_2 I + (c_1 - c_2)\frac{\psi_z \psi_z^T}{\|\psi_z\|^2}, & if \; \psi_z \neq 0 \\ c_1 I, & otherwise \end{cases}, \tag{62}$$

where both $c_1$ and $c_2$ are predefined scales such that $c_2 > c_1 > 0$; this process is called the SDDM, trajectory will be bounded incorporating SDDM information. Since quadrotor dynamics are linear, a reference governor [106] is introduced to maintain safety and stability. Other than LQR and MPC, there exist several receding horizon-based techniques for optimal trajectory planning, as given in Table 2.

**Table 2.** Comparison of the properties of receding horizon trajectory planning techniques. Checks and cross marks indicate whether a feature is available or not.

| Algorithm | Motion Model | | Gradient Estimator | |
|---|:---:|:---:|:---:|:---:|
| | Linear | Nonlinear | Hamiltonian | Gradient |
| Differential Dynamic Programming (DDP) [109] | ✗ | ✓ | ✗ | ✓ |
| Linear Quadratic Regulator (LQR) [110] | ✓ | ✗ | ✗ | ✗ |
| Iterative LQR (iLQR) [111] | ✗ | ✓ | ✗ | ✓ |
| Linear Model Predictive Control (MPC) [112] | ✓ | ✗ | ✗ | ✗ |
| Nonlinear Model Predictive Control (NMPC) [62] | ✓ | ✓ | ✓ | ✗ |
| Constrained Nonlinear Model Predictive Control CGMRES (NMPC-CGMRES) [113] | ✗ | ✓ | ✓ | ✗ |
| Corridor-based Model Predictive Contouring Control (CMPCC) [114] | ✓ | ✗ | ✗ | ✗ |
| Constrained Nonlinear Model Predictive Control Newton (NMPC-Newton) [115] | ✗ | ✓ | ✗ | ✗ |
| Model Preidictive Path Integral Control (MPPI) [116] | ✓ | ✓ | ✗ | ✗ |
| Cross Entropy Method (CEM) [117] | ✓ | ✓ | ✗ | ✗ |

### 8. Solving the Trajectory Planning Problem

As explained in the preceding sections, several constraints (e.g., soft and hard) are imposed to ensure dynamic feasibility, smooth navigation, handling disturbances, etc. Hence, optimal trajectory planning is posed as a constraint optimization problem in most situations. Constraint-based optimization problems are solved in two different ways: adding hard constraints or introducing soft constraints. In general, a constraint-based optimization problem can be formulated as a quadruple, i.e., $P_{constraint} = (c, g1, g2, J)$, where $c$ stands for performance index or cost function, whereas equality and inequality constraints are given by $g1$ and $g2$, respectively. The objective function is given by $J$. In hard-constraint-based formulation, the optimal solution, i.e., **w**, for $P_{constraint}$ is calculated, ensuring all the constraints. In soft constraints formulation, the objective function does not need to satisfy all the constraints, but satisfying those constraints will improve the final **w**. D. Mellinger and V. Kumar [53] took the lead in proposing a successful approach for trajectory generation as a hard-constraint-based optimization approach, i.e., minimum-snap. Subsequently, in [35], the researchers extended the minimum-snap trajectory generation as an unconstrained or soft-constraint-based optimization problem.

When generating trajectories, ensuring a collision-free path is essential. Hence, representing free space in a structured way and imposing obstacle constraints for trajectory generation is a must for safety. Free space can be represented in different ways, such as cubes [26,74], spheres [82,118], and polyhedrons [32]. The intuition of these approaches is to apply path planning through the free space to obtain the intermediate waypoints. Once intermediate waypoints are extracted, the trajectory generation procedure is utilized for retrieving a smooth, feasible, and collision-free trajectory. On the other hand, in [21,25,26], kinodynamic path planning followed by B-spline-based trajectory generation is considered. Most of the works that were proposed for soft-constraint-based trajectory generation formulated optimal trajectory planning as nonlinear optimization problems in which smoothness and safety were introduced as soft constraints. Most of the time gradient-descent-based [50] or gradient-free approaches [37,54] were applied for minimizing the cost of smoothness and safety.

The constraints optimization problem can be designed in either QP or NLP form. In QP, the procedure is to minimize or maximize the objective subject to a set of linear constraints in most situations. On the other hand, nonquadratic programming is used to handle the nonlinear constraints each of which has a unique nature to solve the problem. In general, the QP objective can be described as:

$$\min_{\mathbf{x}} \quad \frac{1}{2}\mathbf{x}^T Q \mathbf{x} + c^T \mathbf{x}$$
$$\text{s.t.} \quad A\mathbf{x} \succeq b, \tag{63}$$

where $A\mathbf{x} \succeq b$ stands for the set of linear inequalities and $Q$ is a positive symmetric matrix. There are various ways to solve QP, including interior point, active set, and gradient projection. In some situations, multiple variables that are to be optimized are integer values; those are solved as MIQP. For example, FASTER [119] used MIQP for safe trajectory planning with aggressive controls [36].

Most of the recent optimal trajectory planning techniques [19,29,30,37] were formulated as gradient-based trajectory optimization (GTO) in which optimization problem was designed as a nonlinear form. The gradient descent is performed with respect to each parametrization index of $\Gamma$ to minimize the difference, i.e., $\Gamma_{i+1} - \Gamma_i$. Hence, $\Gamma_{i+1}$ can be determined by solving the following optimization problem, as given in [50,120].

$$\Gamma_{i+1} = \arg\min_{\Gamma} J(\Gamma_i) + (J(\Gamma) - J(\Gamma_i))^T \bigtriangledown J(\Gamma_i) + \frac{\eta}{2}\|\Gamma - \Gamma_i\|_M^2, \tag{64}$$

where $M$ is a weighting matrix and $\eta$ is a regularization parameter. GTO is rather popular due to its ability to deform ineffability trajectory segments, low memory requirement, and

high throughput. Despite having the listed advantages, GTO cannot avoid the problem of a local minimum. STOMP [54] is one of the early techniques proposed to address the local minimum problem. STOMP is based on the gradient-free technique. However, STOMP is unable to obtain real-time performance. Besides STOMP, the local minimum problem has been addressed by various recent works. However, this remains an open problem to be solved. Zhou [121] proposed a method, i.e., path-guided optimization (PGO), for overcoming local minima problems by generating topologically distinct paths and doing parallel optimization. Furthermore, various solvers can be utilized for solving optimization problems, including BOBYQA [122], L-BFGS [8,123], ACADO [124], SLSQP [125], Proximal Operator Graph Solver (POGS) [126,127], sequential quadratic programming (SQP), and MMA [128]. Shravan et al. [65] proposed a trajectory optimization technique in a distributed setup in which the researchers evaluated their formulation with several solvers. According to their observations, BOBYQA is faster compared to BFGS and SLSQP, while MMA yielded a similar performance to that of BOBYQA. In [129], L-BFGS was proposed for finding the shortest path in real-time; in this research effort, however, L-BFGS does not guarantee optimality, and only feasibility is enforced. Mathematical program with complementarity constraints (MPCC) [130] is yet another proposed method for fast trajectory optimization in real-time. Moreover, Mathieu and Nicolas [131] proposed a SQP-based trajectory generation approach for carrying augmented loads. The intuition behind selecting SQP over other solvers is due to its superlinear convergency and ability to handle nonlinear constraints within milliseconds.

## 9. Conclusions

All in all, we have thoroughly reviewed the trajectory planning problem in the paradigm of plan-based control for multirotor aerial vehicles (MAVs). Such a trajectory planning problem was broken down into a set of subproblems: free-space segmentation, motion model selection, initial waypoints identification, initial trajectory generation, continuous trajectory refinement, and receding horizon trajectory planning. Afterwards, for each subproblem, we examined how previous research has addressed those by presenting and evaluating various approaches to the considered subproblem. Finally, several selected recent approaches were listed (Table 3) according to the listed subproblems we have identified. Furthermore, Table 4 summarizes the key findings of the study, including features and performance. With that, we concluded that the trajectory planning problem can be designed by addressing those subproblems carefully for MAVs.

**Table 3.** The basic building blocks that are encountered in trajectory planning problems as described in the paper.

| Approach | Dynamics Model (Exact \| Empirical Differential Flatness (DF)) | Intermediate Waypoint Selection | Initial Trajectory Generation | Continuous Trajectory Refinement and Solver | Free Space Extraction | Receding Horizon Planning or Controlling |
|---|---|---|---|---|---|---|
| A replanner [121] | DF | Sampling-based topological search | PGO-based B-splines | GTO | ESDF | - |
| A replanner [25] | DF | Kinodynamic-based search | B-splines | EO using QCQP | ESDF | - |
| A replanner [101] | DF | Kinodynamic-based search | Linear quadratic minimum time | Unconstrained QP | [132] | RHC |
| A local replanner [37] | DF | Informed-RRT* | Continuous time polynomial | BFGS | ESDF | - |
| Teach-repeat-replan [14] | DF | - | Minimum-jerk | Elastic band optimization | Convex Cluster | - |
| Fast planner [19] | DF | A* kinodynamic search | B-splines | NLopt [133] | ESDF | GTC |
| Areplanner [21] | DF | B-spline kinodynamic search | EO | ~QCQP | TSDF | - |
| Chomp [50] | DF | - | CHOMP | Functional gradient [120] | ESDF | CHOMP |
| EGO-Planner [134] | DF | A* | Uniform B-spline | T-NEWTON [135] | ESDF | - |
| A replanner [26] | DF | Fast marching-based search | Bernstein polynomial | Mosek [136] | TSDF | - |
| A safe trajectory generator [82] | DF | RRG combined with A* | Piece-wise polynomials | QCQP | KD-tree | GTC |
| ILQR [91] | Exact | line search | Iterated LQR Smoothing | Iterated LQR Smoothing | - | - |
| Monocular visual-inertial fusion [28] | Exact | A* | VINS | Gradient-based | TSDF | GTC |
| A replanner [78] | DF | RRT* | Uniform-Bspline | MMA and BFGS | OctoMap | GTC |
| Safe flight corridors [32] | DF | JPS | Minimum-span | Constrained QP | SFC | RHC |
| SDDM [108] | Empirical | Piece-wise-linear path | SDDM | SDDM | Constrained QP | - |
| Faster [119] | DF | JPS | Cubic Bézier curve | MIQP using Gurobi [137] | SFC | - |
| CMPCC [114] | Empirical | - | CMPCC | OSQP [138] | SFC | RHC |
| Relative trajectory tracking control [61] | Empirical | - | NMPC | ACADO [124] | - | MHE |
| A trajectory tracker [139] | Empirical | - | NMPC | SQP | - | RHC |
| A replanner [74] | DF | A* | Multi-segment polynomial | OOQP [140] | OctoMap | GTC |
| A replanner [35] | Exact | RRT* | Minimum-span | Unconstrained QP | OctoMap | GTC |
| MADER [141] | DF | MINVO basis [142] | Uniform B-spline | Augmented Lagrangian [143] | Outer polyhedral | - |
| SOS programming [41] | DF | Piece-wise linear path | Piece-wise-polynomial | MIQP using Mosek | IRIS | - |
| A replanner [144] | DF | Nonuniform kinodynamic search | Uniform B-spline | Constrained QP | ESDF | RHC |
| A trajectory tracker [62] | Empirical | Uniform B-spline | NMPC | CasADi [145] with Ipopt [146] | ESDF | PID |

**Table 4.** The comparison of contrasts features and performance of selected different approaches for trajectory planning.

| Approach | Specific Features | Performance Indicates |
|---|---|---|
| A replanner [121] | A path-guided optimization (PGO) approach to address infeasible local minima problems, not limited to a specific use | Computation time (≈15 ms), perform aggressive maneuvers |
| A replanner [25] | A dynamically feasible time parameterized trajectory generation to overcome the limitation of the greedy search, not limited to a specific use | Computation time (≈30 ms), limited maneuvering capabilities |
| A replanner [101] | Searches for smooth, minimum-time trajectories using a set of short-duration motion primitives, not limited to a specific use | Computation time (≈15 ms), limited maneuvering capabilities |
| Teach-repeat-replan [14] | Generate safe local trajectories (smooth, safe, and kinodynamically feasible) to avoid moving obstacles, infrastructure inspection, aerial transportation, and search-and-rescue | Computation time (≈15 ms), perform aggressive maneuvers |
| Fast planner [19] | Kinodynamic feasible and minimum-time trajectory generation in the discretized control space, not limited to a specific use | Computation time (≈5 ms), perform aggressive maneuvers |
| EGO-Planner [134] | A Euclidean signed distance field (ESDF)-free gradient-based planner, not limited to a specific use | Computation time (≈2 ms), extreme maneuvering capabilities |
| ILQR [91] | LQR smoothing to compute a locally optimal feedback control policy, can work with nonlinear dynamics and nonquadratic cost, limited to a specific use | Computation time (≈3 s), limited maneuvering capabilities |
| Monocular visual-inertial fusion [28] | A monocular visual-inertial navigation system (VINS), consisting only an inertial measurement unit (IMU) and a camera. VINS supports self-extrinsic calibration | Computation time (≈30 ms), limited maneuvering capabilities |
| Safe flight corridors (SFC) [32] | The SFC is a set of overlapping convex polyhedra that represent free space and provide a connected path for the robot to reach its goal. | Computation time (≈100 ms), extreme maneuvering capabilities |
| SDDM [108] | A control policy for MAVs systems that uses ellipsoidal trajectory bounds defined by a quadratic state-dependent distance metric. SDDM behavior is adapted to the geometry of the local environment. | Computation time (≈100 ms), limited maneuvering capabilities |
| FASTER [119] | FASTER guarantees safety without compromising speed by having a safe backup trajectory, and MIQP is used to allocate trajectory intervals | Computation time (≈15 ms), extreme maneuvering capabilities |
| MADER [141] | MADER uses the MINVO basis to generate trajectories through free space more effectively than Bernstein or B-Spline bases in obstacle-dense environments. | Computation time (≈10 ms), extreme maneuvering capabilities |
| SOS programming [41] | A sums-of-squares (SOS) programming approach that ensures the entire piece-wise-polynomial trajectory is collision-free using convex constraints. | Computation time (≈60 ms), limited maneuvering capabilities |
| A trajectory tracker [62] | Nonlinear model predictive control (NMPC) with multiple shooting is used to predict the optimal control policy at each iteration. | Computation time (≈60 ms), limited maneuvering capabilities |
| Residual dynamics [147] | A learning-based technique using Sparse Gaussian Process Regression is proposed to reduce the residual dynamics between high-level planning and low-level controlling | Computation time (≈20 ms), extreme maneuvering capabilities |
| A replanner [8] | A continuous optimization-based method for refining the reference trajectory to move it out of obstacle-occupied space in the global phase. | Computation time (≈15 ms), extreme maneuvering capabilities |

Our hypothesis was to investigate trajectory planning for multirotor aerial vehicles (MAVs) in the plan-based control paradigm, focusing on analytical approaches rather than fully learnable approaches, such as machine learning and deep learning. In future work, we plan to investigate other types of trajectory planning approaches, such as those based on imitation learning and inverse reinforcement learning. We also plan to conduct simulated and real-world experiments to compare the performance of the considered approaches in various conditions, such as high-dense and less-dense environments, and with static and dynamic obstacles. These considerations were not included in this work, as we focused on the theoretical aspects of trajectory planning.

**Author Contributions:** G.K. proposed the structure of the paper and carried out a comprehensive review in the field of motion planning targeting multirotor aerial vehicles. G.K. wrote the manuscript in consultation with A.K. A.K. made a critical reviewing the manuscript and final approval for publication. All authors have read and agreed to the published version of the manuscript.

**Funding:** This research received no external funding.

**Conflicts of Interest:** The authors declare no conflict of interest.

## Abbreviations

The following abbreviations are used in this manuscript:

| | |
|---|---|
| OCP | Optimal Control Problem |
| OCPs | Optimal Control Problems |
| MHE | Model Horizon Estimation |
| NMPC | Nonlinear Model Predictive Control |
| LQR | Linear Quadratic Regulator |
| iLQR | Iterative Linear Quadratic Regulator |
| MPC | Model Predictive Control |
| DDP | Differential Dynamic Programming |
| NLP | Nonlinear Programming |
| QP | Quadratic Programming |
| MIQP | Mixed Integer Quadratic Programming |
| CBFs | Control Barrier Functions |
| SDDM | State-dependent Distance Metric |
| CMPCC | Corridor-based Model Predictive Contouring Control |
| RRG | Rapidly-exploring Random Graph |
| IRIS | Iterative Regional Inflation by Semi-definite Programming |
| SFC | Safe Flight Corridor |
| JPS | Jump Point Search |
| GTO | Gradient-based Trajectory Optimization |
| SQP | Sequential Quadratic Programming |
| MPCC | Mathematical Program with Complementarity Constraints |
| ESDF | Euclidean Signed Distance Field |
| PGO | Path-guided Optimization |
| LTI | Linear Time Invariant |
| TOPP | Time-Optimal Parameterization of a given Path |
| CHOMP | Covariant Hamiltonian Optimization for Motion Planning |
| MAVs | Multirotor Aerial Vehicles |
| MAV | Multirotor Aerial Vehicle |
| UAVs | Unmanned Aerial Vehicles |
| UAV | Unmanned Aerial Vehicle |
| LQG | Linear Quadratic Gaussian |
| KF | Kalman Filter |
| EO | Elastic Optimization |
| QCQP | Quadratically Constrained Quadratic Programming |
| RHC | Receding Horizon Control |
| BFGS | Broyden–Fletcher–Goldfarb–Shanno |
| TSDF | Truncated Signed Distance Field |

| | |
|---|---|
| PRM | Probabilistic Road Map |
| GTC | Geometric Tracking Control |
| DoF | Degree of Freedom |

**Symbols**

| | |
|---|---|
| $\mathbf{x}$ | State vector and its derivative is denoted as $\dot{\mathbf{x}}$. Term $\mathbf{x}^+$ depicts the next state given the current state $\mathbf{x}$, and term $\mathbf{x}_k$ denotes discrete state at time $t$ equals $k$ |
| $\mathbf{u}$ | Control input. The term $\mathbf{u}*$ denoted as the optimal control inputs |
| $\mathbf{p}$ | Position (m) in $\mathbb{R}^3$ and its derivative is denoted as $\dot{\mathbf{p}}$. $\mathbf{p}_*, * \in x, y, z$, stands for position alone * component |
| $p$ | $d$th-order polynomial, which is a function of time. Term $\dot{p}(t)$, $\ddot{p}(t)$ (or $p(t)^{(1)}$, $p(t)^{(2)}$) denote the higher order derivatives of $p(t)$ |
| $\lambda$ | Polynomial coefficients, e.g., $p(t) = \lambda_d t^d + \ldots + \lambda_1 t + \lambda_0$, $t \in [0, dt]$, where $d$ is the order of the polynomial |
| $\mathbf{v}$ | Velocity (m/s) in $\mathbb{R}^3$ and its derivative is denoted as $\dot{\mathbf{v}}$. $\mathbf{v}_*, * \in x, y, z$, stands for velocity alone * component |
| $\omega$ | Angular velocity (rad/s) in $\mathbb{R}^3$ and its derivative is denoted as $\dot{\boldsymbol{\omega}}$ |
| $\psi$ | Orientation is represented as quaternion in $\mathbb{R}^3$ and its derivative is denoted as $\dot{\boldsymbol{\psi}}$. $\boldsymbol{\psi}_*, * \in x, y, z$, stands for orientation alone * component |
| $\mathbf{f} = [f_1, f_2, f_3, f_4]^T$ | System input or total trust that is applied for each of the motors in N (Newton) |
| $\mathbf{f}_d$ | Discrete system dynamics |
| $\mathbf{f}_c$ | Continuous system dynamics |
| $\delta$ | Euler or Runge Kutta discretization time step |
| $\mathbf{z}$ | System output |
| $(q)$ | Apices $^{(q)}$ stipulates the $q$th derivative, for example $\mathbf{z}^{(q)}$ |
| $C$ | Configuration space that can be one of these: $C_{free}$, $C_{obs}$, $C_{unknown}$, and $C_{unknown}$ |
| $d$ | Order of polynomial |
| $\Gamma$ | Initial trajectory; the optimal trajectory is defined as $\Gamma^*$, trajectory derivatives are defined as $\dot{\Gamma}$ and $\ddot{\Gamma}$, and trajectory is a function of time, i.e., $\Gamma_T(t)$ |
| $\xi$ | Regularization parameter |
| $c$ | Formulation of cost function, where $c(\cdot), \cdot$ denotes the inputs |
| $A$ | $H$ representation of polytope, i.e, $A^T \mathbf{p} = b$ |
| $\mathbf{w}$ | The optimal estimation for states and/or controls after minimizing given cost function |
| $g$ | Equality constraints are denoted by $g_1(\mathbf{w})$, whereas inequality constraints are denoted by $g_2(\mathbf{w})$ |

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
