# Peer review of "Survey on Motion Planning for Multirotor Aerial Vehicles in Plan-Based Control Paradigm"

_remotesensing, doi:10.3390/rs15215237_

Round 1

Reviewer 1 Report

Comments and Suggestions for Authors

Dear Authors;

Firstly I congratulate your paper. 

Some areas need to be checked and corrected. Please find them as follows;

Comment - 1

(Line-25)

The word "areal" should be changed to "aerial"

Comment - 2

(Line-93)

Spell out the full term at its first mention, indicate its abbreviation in parenthesis, and use the abbreviation from then on.

e.g.

Multirotor Aerial Vehicle (MAV)

Comment – 3

Figure 2 The yellow-colored letters are not readable. They should be rewritten in a larger format. Please correct the size of the yellow-colored sentences.

Comment - 4

(Line-104)

Spell out the full term at its first mention, indicate its abbreviation in parenthesis, and use it from then on.

Degrees of Freedom (DOF)

Comment – 5

(Line-142, 146)

Fatness or Flatness

Comment – 6

Figure 4 is too small to recognize. It should be bigger.

Comment – 7

(Line-108)

In aeronautical science, when describing an air vehicle's position, we should provide three references for waypoints. These are longitude, latitude, and altitude.

The authors should make the corrections for providing information regarding waypoints and positions of the MAV.

Comment – 8

(Line-705)

The sentence should be corrected.

Comment – 9

(Line-851)

The last Entrance Date should be written for Reference #1

Comment – 10

(Line-852)

The last Entrance Date should be written for Reference #1

Comment – 11

(Line-1083)

There is a misspelling in Reference #118.

“https://doi.org" was written two times by mistake.

Best Regards

Comments on the Quality of English Language

The quality of the English is sufficient and acceptable.

Author Response

Please, see attachment.

Reviewer 2 Report

Comments and Suggestions for Authors

The paper focuses on the critical issue of optimal motion planning for Multirotor Aerial Vehicles (MAVs), such as drones, in dynamic and obstacle-rich environments. It explores the challenges associated with achieving efficient and optimal motion planning in scenarios where environmental conditions change rapidly and unpredictable events can occur. The paper categorizes the optimal motion planning problem into subproblems, such as motion modeling, waypoint identification, trajectory generation, refinement, and receding horizon planning. It reviews the state-of-the-art techniques and approaches used to address each of these subproblems and presents a timeline of prominent results from 2010 to 2022 in the field. The paper aims to provide insights into how researchers have been tackling these challenges to enable robust navigation for MAVs in various domains.

Recommendations:

Research Question or Hypothesis: The paper lacks a specific research question or hypothesis. It would be beneficial to include a clear and concise statement of the research objectives to provide a focused direction for the study.

Real-World Examples: To emphasize the significance of the problem, consider including real-world examples or statistics demonstrating the practical importance of optimal motion planning for MAVs in various applications.

Methodology Details: Provide more specific details about the techniques or algorithms used in each of the identified subproblems, such as motion modeling, waypoint identification, and trajectory generation. This will enhance the understanding of the methods discussed.

Conclusion: Consider adding a section on future research directions to suggest potential areas for further investigation in the field of MAV trajectory planning.

Language and Style: Review and simplify lengthy sentences for improved readability. Conduct thorough proofreading to eliminate grammatical errors and typos throughout the paper.

Original Contribution: Emphasize the paper's original contributions or insights, if applicable, to highlight its unique value in the context of existing research.

Comments on the Quality of English Language

Review and simplify lengthy sentences for improved readability. Conduct thorough proofreading to eliminate grammatical errors and typos throughout the paper.

Author Response

Please, see attachment.

Reviewer 3 Report

Comments and Suggestions for Authors

The author discusses various aspects related to trajectory planning for MAVs (Unmanned Aerial Vehicles), including the importance of considering a set of constraints (both hard and soft constraints) to ensure smooth and safe navigation, as well as methods for solving constraint optimization problems.

The autor also mentions different techniques and algorithms used in trajectory planning, such as the Linear Quadratic Regulator (LQR), Model Predictive Control (MPC), and the Cross Entropy Method (CEM). It also discusses ways of representing free space and imposing constraints for collision avoidance.

Furthermore, the autor focuses on optimization methods, such as Quadratic Programming (QP) and Non-Linear Programming (NLP), in the context of trajectory planning. It is noted that these problems can have multiple solutions, including solutions with hard or soft constraints.

In line 765 there is a sign ?. Maybe is number 3 there.

In conclusion, the paper provides an overview of the issues related to trajectory planning for MAVs and lists some of the methods and technologies used in research in this field.

Author Response

Please, see attachment.

Reviewer 4 Report

Comments and Suggestions for Authors

This report on trajectory planning for Micro Aerial Vehicles (MAVs) provides a detailed and informative overview of the key challenges and approaches in this field. The authors have successfully highlighted the complexities involved in MAV trajectory planning, including the selection of motion models, waypoint identification, and dealing with sudden disturbances. They've also elucidated the significance of continuous trajectory refinement and receding horizon planning, demonstrating the importance of ensuring dynamic feasibility and safety in the context of real-time operation.

The report's clear structure and explanations make it a valuable resource for both researchers and practitioners in the field of robotics and aerial vehicles. It effectively outlines the various optimization methods and solvers used in trajectory planning, offering readers a comprehensive understanding of the subject. The inclusion of tables and comparisons adds to the report's clarity and makes it a useful reference for those seeking insights into state-of-the-art techniques for MAV trajectory planning.

In conclusion, this report serves as an insightful guide to understanding the nuances of trajectory planning for MAVs, making it a valuable contribution to the field of autonomous aerial systems.

Author Response

Please, see attachment.

Reviewer 5 Report

Comments and Suggestions for Authors

(1) It is suggested to list the advantages, disadvantages and applicable scenarios of motion planning methods. It will be more instructive

(2) It is suggested that sections 4.7 and 4.8 should be trajectory optimization rather than initial trajectory generation.

(3) Line 409: "in Fig.5" should be "in Fig.7".

(4) It is recommended that the first column of Table 3 should give the name of the approach rather than the reference number.

Author Response

Please, see attachment.

Round 2

Reviewer 2 Report

Comments and Suggestions for Authors

Accept in the present revised version form.

Reviewer 5 Report

Comments and Suggestions for Authors

The revised manuscript can be accepted.